# GOT-Edit: Geometry-Aware Generic Object Tracking via Online Model Editing

**Shih-Fang Chen**[1]    **Jun-Cheng Chen**[2]    **I-Hong Jhuo**[3]    **Yen-Yu Lin**[1]

[1]Department of Computer Science, National Yang Ming Chiao Tung University
[2]Academia Sinica    [3]Microsoft AI

## Abstract

Human perception for effective object tracking in 2D video streams arises from the implicit use of prior 3D knowledge and semantic reasoning. In contrast, most generic object tracking (GOT) methods primarily rely on 2D features of the target and its surroundings, while neglecting 3D geometric cues, making them susceptible to partial occlusion, distractors, and variations in geometry and appearance. To address this limitation, we introduce GOT-Edit, an online cross-modality model editing approach that integrates geometry-aware cues into a generic object tracker from a 2D video stream. Our approach leverages features from a pre-trained Visual Geometry Grounded Transformer to infer geometric cues from only a few 2D images. To address the challenge of seamlessly combining geometry and semantics, GOT-Edit performs online model editing. By leveraging null-space constraints during model updates, it incorporates geometric information while preserving semantic discrimination, yielding consistently better performance across diverse scenarios. Extensive experiments on multiple GOT benchmarks demonstrate that GOT-Edit achieves superior robustness and accuracy, particularly under occlusion and clutter, establishing a new paradigm for combining 2D semantics with 3D geometric reasoning for generic object tracking. The project page is available at `https://chenshihfang.github.io/GOT-EDIT`.

## 1 Introduction

Generic object tracking (GOT) (Bhat et al., 2019; Li et al., 2019; Javed et al., 2022) aims to track an arbitrary user-specified target object, identified by its initially bounding box in the first frame, and to predict the locations of this target in subsequent frames. However, learning a robust tracker from limited visual information remains a significant challenge, especially in adverse conditions like partial occlusion, cluttered scenes with distractors, and significant object deformations.

Most contemporary GOT trackers are trained on 2D datasets, e.g., (Muller et al., 2018; Fan et al., 2019; Huang et al., 2019; Peng et al., 2024). As a result, their 2D-based representations limited their ability to reason about contextual relationships between a target and its surroundings, such as distinguishing a target under partial occlusion or separating it from background distractors. In contrast, incorporating 3D information provides geometric cues for object boundaries, enabling more precise reasoning to mitigate challenges such as partial occlusion and inter-object discrimination.

Although several studies (Tan et al., 2025a;b; Chen et al., 2025b; Feng et al., 2025; Hu et al., 2025; Xu et al., 2025b; Zhang et al., 2024a) have attempted to leverage 3D information for enhanced tracking, they often rely on additional 3D data, such as objects represented in RGB-D or backgrounds in point clouds. This reliance is impractical, as GOT is primarily performed on 2D video streams. Humans, by contrast, can track targets from the background, near or far, even when observing only 2D videos or single images. This is because our prior 3D knowledge allows for perception that extends beyond the flat image plane (Koch et al., 2018; Gregory, 1997).

Emerging techniques in geometric 3D vision (Wang et al., 2024; 2025a; Zhang et al., 2025; Wang et al., 2025b; Yang et al., 2025) offer a promising direction for advancing GOT. Among these, we adopt the Visual Geometry Grounded Transformer (VGGT) (Wang et al., 2025a) for its strong performance and generalization, in alignment with the GOT objectives. Given one or a few 2D images as input, VGGT learns features for camera pose, point map, and depth estimation. While

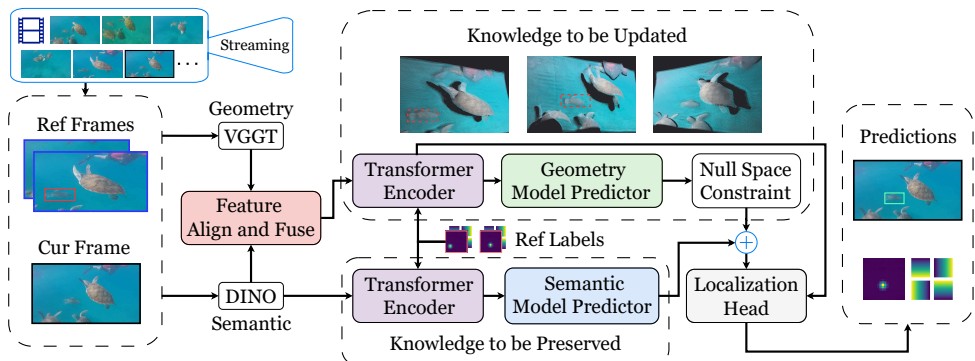

Figure 1: **The GOT-Edit Framework.** GOT-Edit facilitates the understanding of 3D geometry to aid generic object tracking from 2D streaming inputs. It predicts semantic and geometric model weights concurrently to incrementally adapt the tracking model. Through online model editing, it ensures geometry-aware, semantic-preserving updates to the tracking model. The solid red box marks the ground-truth target in the input reference frames. The dashed red boxes indicate these same annotations utilized for the online knowledge update within the geometry branch. The green box represents the final predicted tracking result.

VGGT has shown effectiveness in point tracking (Wang et al., 2025a; Karaev et al., 2024), perception from 2D semantics remains essential for GOT. This is because point tracking operates at the pixel level and does not require an understanding of object semantics, whereas a robust GOT tracker benefits from both geometric and semantic information.

While geometric information is potentially beneficial for GOT, effectively balancing its contribution with crucial or even dominant semantic information remains a key challenge. As evidenced by our later experiment, a naive fusion strategy improves geometry attributes in tracking but degrades semantic attributes. To address this issue, we propose an online model editing technique that better integrates 3D geometric features (Wang et al., 2025a) with 2D semantic features (Oquab et al., 2023) from only 2D streaming inputs. Our approach is inspired by the *null-space model editing* of AlphaEdit (Fang et al., 2025), which integrates new knowledge into a trained model while preserving existing knowledge through null-space constraints. However, AlphaEdit performs offline editing, whereas GOT requires online updates to handle dynamically varying targets and backgrounds in seen and unseen scenarios. To bridge this gap, we develop an online editing technique that enables a tracker to adaptively complement 2D semantics with 3D geometric features.

As illustrated in Figure 1, our system begins by extracting both semantic and geometric features from the current and reference frames. These features are then aligned and fused to create an enriched representation, which serves as new knowledge for online tracker adaptation. Built upon the ToMP (Mayer et al., 2022), our approach employs two model predictors: one for the semantic branch and one for the geometric branch. These predictors generate the model weights for the localization head. During tracking, the reference labels, which provide correspondences between the reference frames and serve as few-shot examples of previously predicted and observed information, are dynamically updated to guide the tracker toward the target object. This process guides the model predictors to forecast model weights for the current frame in an online manner. Namely, the semantic model predictor estimates the semantic weights, while the geometry predictor generates complementary weights. A null-space constraint is applied before combining these two sets of weights to preserve the semantic information. Finally, the combined model weights are used by the localization head to localize the target in the current frame.

Our main contributions are threefold. First, we integrate semantic and geometric knowledge into generic object tracking without relying on additional 3D input data. This integration enriches 2D tracking with geometry-aware reasoning, strengthening target discrimination in complex environments. Second, we propose an online model editing method with a null-space constraint, which adaptively incorporates additional 3D geometric knowledge into GOT without degrading the dominant semantic features. Finally, extensive experiments on multiple benchmarks validate the effectiveness of our approach, demonstrating that it unlocks most of the geometric knowledge lacking in existing 2D trackers, resulting in superior performance.

## 2 RELATED WORK

**Generic Object Tracking.** Existing methods for Generic Object Tracking (GOT) task are typically derived from two pipelines (Javed et al., 2022): matching-based trackers and tracking-by-detection trackers. The matching-based paradigm formulates tracking as a similarity learning task followed by matching (Bertinetto et al., 2016; Li et al., 2018; Guo et al., 2020; Xu et al., 2020; Voigtlaender et al., 2020; Yu et al., 2020; Zhang et al., 2020; Yan et al., 2021a; Cheng et al., 2021; Chen et al., 2021; Ye et al., 2022; Guo et al., 2022; Cai et al., 2023; Gao et al., 2022; He et al., 2023; Zhou et al., 2023; Li et al., 2023; Chen et al., 2023; Jinxia et al., 2024; Bai et al., 2024; Shi et al., 2024; Cai et al., 2024; Song et al., 2023; Wu et al., 2023; Zhao et al., 2023; Zhang et al., 2024b; Xie et al., 2024; Zhu et al., 2025). These methods focus on training a deep network to learn a function that can distinguish and match a template of the object to a search region in the current frame. This trained network is then used for tracking. Recent matching-based trackers (Guo et al., 2025; Xu et al., 2025a; Li et al., 2025; Kang et al., 2025; Xie et al., 2025) further improve their robustness by propagating chronological contextual information from predicted hidden states.

Another paradigm, tracking-by-detection, frames generic object tracking as an online detection task (Bolme et al., 2010; Henriques et al., 2012; Nam & Han, 2016; Kiani Galoogahi et al., 2017; Yao et al., 2018; Lukezic et al., 2017; Danelljan et al., 2017; 2016; Nai & Chen, 2023; Jia et al., 2024). Recent trackers under this paradigm employ a model predictor that generates a target-specific tracking model from paired reference images and labels, allowing more accurate object localization in the current frame (Bhat et al., 2019; Mayer et al., 2022; Chen et al., 2025a). The model predictor is dynamically updated for each incoming frame by referring to previous tracking results and hence enhances the tracker's robustness and adaptivity. A separate localization head then uses this updated model to pinpoint the target.

Despite progress in the above two paradigms, they remain limited by their reliance solely on two-dimensional spatial and structural knowledge. To overcome this, our method integrates 2D semantic information with 3D geometric features, enabling a 2D tracker to exploit 3D geometry information through online tracking model editing.

**3D Features for Tracking.** Existing trackers that utilize 3D features fall into two primary categories: those that augment RGB images with additional modalities (RGB+X) (Yan et al., 2021b; Yang et al., 2022; Zhu et al., 2023; Hou et al., 2024; Cao et al., 2024; Tan et al., 2025a;b; Chen et al., 2025b; Feng et al., 2025; Hu et al., 2025) and those that operate directly on point cloud data (Wu et al., 2024; Nie et al., 2024; Liu et al., 2024a; Zhang et al., 2024a; Seidenschwarz et al., 2024; Xu et al., 2025b). These approaches require auxiliary inputs during tracking, such as pre-computed depth maps or scene point clouds, which are generally unavailable in real-world scenarios where scenes and objects may be arbitrary and even previously unseen. Another line of research (Doersch et al., 2022; Harley et al., 2022; Doersch et al., 2023; Wang et al., 2023; Karaev et al., 2024; Kim et al., 2025), known as point tracking, explores tracking any pixel. Recent extensions (Xiao et al., 2025; Lai & Vedaldi, 2025; Rajič et al., 2025; Wang et al., 2025a; Harley et al., 2025) incorporate 3D information for point tracking.

Unlike these methods, our tracker adaptively integrates 3D geometric knowledge with 2D semantic knowledge for GOT through online model editing. Specifically, we embed VGGT (Wang et al., 2025a) into a 2D tracker, where a sequence of RGB frames is used to derive complementary 3D information. While geometric features from VGGT have proved effective for point tracking (Wang et al., 2025a; Karaev et al., 2024), our method departs from this line of work by embedding these features into a 2D GOT tracker via model editing, thereby establishing a direct connection between 3D geometric representations and object-level semantics for tracking. In this way, our formulation operates directly on RGB streams and extracts geometric cues from them, yielding a geometry–aware and semantics–preserving GOT formulation that matches the intrinsic nature of the task and aligns with the way human observers infer scene structure from two-dimensional imagery.

## 3 METHOD

Geometry inferred from 2D visual streams benefits GOT by enabling a tracker to move beyond flat representations, but it must be balanced with semantic knowledge. Driven by this insight, we aim to enhance tracking with geometry-aware reasoning while preserving semantic discrimination.

In the following sections, we introduce null-space model editing in AlphaEdit and explain how it links geometry and semantics (Section 3.1.1). We then justify the track-by-detection paradigm as a natural fit for model editing (Section 3.1.2). Finally, we provide a step-by-step description of our pipeline, highlighting our online model editing approach and objective function (Section 3.2).

## 3.1 PRELIMINARY

### 3.1.1 NULL-SPACE CONSTRAINED KNOWLEDGE EDITING

Model editing updates the knowledge stored in a model by adjusting its learned weights. Among existing model editing algorithms, we adopt the AlphaEdit (Fang et al., 2025) because it excels at fusing unbalanced features while avoiding catastrophic forgetting. AlphaEdit treats the feed-forward network (FFN) as a linear associative memory, where input features serve as keys and are mapped to output features through model parameters $\mathbf{W} \in \mathbb{R}^{d_b \times d_a}$:

$$\mathbf{V} = \mathbf{W}\mathbf{K}, \text{ where } \mathbf{K} = [\mathbf{k}_1 \mid \mathbf{k}_2 \mid \ldots \mid \mathbf{k}_u] \in \mathbb{R}^{d_a \times u} \text{ and } \mathbf{V} = [\mathbf{v}_1 \mid \mathbf{v}_2 \mid \ldots \mid \mathbf{v}_u] \in \mathbb{R}^{d_b \times u}. \quad (1)$$

In Eq. 1, $u$ is the number of features to be updated, $d_a$ and $d_b$ are the dimensions of the respective FFN layers, and $\mathbf{k}_i \in \mathbb{R}^{d_a}$ and $\mathbf{v}_i \in \mathbb{R}^{d_b}$ jointly represent the $i$-th key-value pair.

One representative optimization objective for model editing is defined by:

$$\mathbf{\Delta} = \arg\min_{\tilde{\mathbf{\Delta}}} \left( \left\| (\mathbf{W} + \tilde{\mathbf{\Delta}})\mathbf{K}_1 - \mathbf{V}_1 \right\|^2 + \left\| (\mathbf{W} + \tilde{\mathbf{\Delta}})\mathbf{K}_0 - \mathbf{V}_0 \right\|^2 \right), \quad (2)$$

where $\mathbf{K}_0$ and $\mathbf{V}_0$ represent originally learned knowledge, while $\mathbf{K}_1$ and $\mathbf{V}_1$ encode newly introduced knowledge. This objective seeks an optimal perturbation $\mathbf{\Delta}$, obtained by optimizing over candidate perturbations $\tilde{\mathbf{\Delta}}$, to edit the model to account for both original and new knowledge.

In practice, new edits often degrade performance on the learned knowledge, as original associations are disrupted. AlphaEdit addresses this by introducing a null-space constraint: the perturbation $\mathbf{\Delta}$ is required to lie in the *null space* of $\mathbf{K}_0$, i.e., $\mathbf{\Delta}\mathbf{K}_0 = \mathbf{0}$. It follows that

$$(\mathbf{W} + \mathbf{\Delta})\mathbf{K}_0 = \mathbf{W}\mathbf{K}_0 = \mathbf{V}_0. \quad (3)$$

This additional constraint ensures preservation of the learned knowledge when adapting the model to new knowledge. Thus, AlphaEdit is highly suitable for our proposed GOT-Edit, where dominant 2D semantic features serve as the knowledge to be preserved, while auxiliary 3D geometric features represent the newly introduced knowledge. Specifically, the tracker predicts the semantic model weights online and the perturbation weights from 3D features concurrently. These geometry-aware perturbation weights are projected into the null space of the semantic knowledge to preserve semantics. The semantic weights and the projected perturbation weights are then combined, enabling a dedicated integration of both semantic and geometric information for object tracking.

### 3.1.2 TRACK-BY-DETECTION PARADIGM

The track-by-detection paradigm (Henriques et al., 2012; Javed et al., 2022) forms the foundational framework for our GOT-Edit tracker. In this paradigm, a tracker predicts a target-specific tracking model (or filters), updates it dynamically online, and employs this model to localize the target in the current frame, thereby performing tracking by detection in an online manner.

Recent trackers (Bhat et al., 2019; Mayer et al., 2022; Chen et al., 2025a) in this paradigm employ a model predictor to generate the weights $\mathbf{W}$ for the localization head of the tracker. The weights are applied to the current frame features $z_{cur}$ through convolution or matrix multiplication to produce a classification score map $p$, which highlights the target's location in the current frame at the feature resolution:

$$p = \mathbf{W} * z_{cur}. \quad (4)$$

Our GOT-Edit framework aims to adapt the $\mathbf{W}$ with the new knowledge through online model editing. As the formulation in Eq. 4 shares a similar form to the linear equation of AlphaEdit, it allows GOT-Edit with AlphaEdit-like online model editing to make the fused knowledge semantics-preserved and geometry-aware, thereby improving the generalization of the tracker.

### 3.2 GOT-EDIT

By combining 2D semantic understanding with 3D geometric reasoning, GOT-Edit enables trackers to preserve semantic knowledge while adaptively incorporating geometric cues. In the following, we first present the pipeline that fuses semantics and geometry for GOT, and then describe the model-editing mechanism that regulates their interaction and ensures coherent cooperation between semantic and geometric modalities.

**Feature Extraction.** Given the reference frames (from previous frames) and the current frame (to be localized), we extract their semantic features (Oquab et al., 2023), $v_{ref}^s \in \mathbb{R}^{C \times H \times W}$ and $v_{cur}^s \in \mathbb{R}^{C \times H \times W}$, and geometric features (Wang et al., 2025a), $v_{ref}^g \in \mathbb{R}^{C' \times H' \times W'}$ and $v_{cur}^g \in \mathbb{R}^{C' \times H' \times W'}$. Note that two reference frames are used, but only one is shown here for brevity.

**Alignment and Fusion.** The geometric features are aligned to match the dimensionality and resolution of semantic features using a convolutional network $Align(\cdot)$ and then fused with semantic features via a gating mechanism:

$$F_{ref} = v_{ref}^s + m_{ref} \odot Align(v_{ref}^g) \quad \text{and} \quad F_{cur} = v_{cur}^s + m_{cur} \odot Align(v_{cur}^g), \quad (5)$$

where $\odot$ denotes point-wise multiplication; $m_{ref} \in [0,1]^{C \times H \times W}$ and $m_{cur} \in [0,1]^{C \times H \times W}$ are spatial gating masks predicted from the paired semantic and geometric features via a lightweight convolution and a sigmoid function, for both of the reference and current frames, respectively.

**Model Predictor.** After fusing the semantic and geometric features, they are spatially concatenated with positional encodings and fed into the model predictor, a Transformer encoder-decoder (Mayer et al., 2022; Carion et al., 2020). The encoder $T_{enc}$ performs feature interaction, i.e.,

$$(z_{ref}, z_{cur}) = T_{enc}([F'_{ref}, F_{cur}]), \quad \text{where} \quad F'_{ref} = F_{ref} + (L_{ref} \cdot e_{fg}). \quad (6)$$

In Eq. 6, $L_{ref}$ denotes the reference labels from past predictions, which indicate the correspondence of the target coordinates to the reference frame. $e_{fg}$ is a learned foreground embedding (Mayer et al., 2022), and the operator $\cdot$ denotes point-wise multiplication with broadcasting.

The resulting features from Eq. 6, together with the learned foreground embedding $e_{fg}$ serving as the query, are fed into a Transformer decoder (Mayer et al., 2022; Carion et al., 2020) $T_{dec}$, which generates the weights $\Delta \in \mathbb{R}^C$ of the localization head via:

$$\Delta = T_{dec}([z_{ref}, z_{cur}], e_{fg}). \quad (7)$$

**Localization Head.** The fused features of the current frame are then passed to the updated localization head for target localization:

$$p = \Delta * z_{cur}. \quad (8)$$

It is important to note that $F'_{ref}$ in Eq. 6 provides important information to differentiate the spatial and geometric properties of the target from the background in the reference frames and can serve as few-shot examples to guide target prediction in the current frame.

**Online Model Editing.** Integrating 3D features enhances GOT by enabling geometric reasoning. However, their influence must be carefully balanced with semantic information, as naive fusion can degrade semantic discrimination, as shown in Table 5. Semantic cues remain the primary signal for distinguishing the target from distractors, whereas geometric cues provide complementary robustness. GOT-Edit therefore performs online model editing that projects geometry-induced perturbations into the null space of semantic features, resulting in an asymmetric interaction that preserves semantic knowledge while still leveraging geometric information.

Specifically, we develop a mechanism that preserves semantic knowledge while incorporating geometric cues by reformulating Eq. 8 as follows:

$$p = (\mathbf{W}_{sem} + \mathbf{\Delta}') * z_{cur}, \quad (9)$$

where $\mathbf{W}_{sem} \in \mathbb{R}^C$ denotes the semantic weights, obtained by passing semantic features through the ***semantic model predictor***. This process is analogous to those described in Eqs. 6 and 7, but uses only semantic features as input. $z_{cur} \in \mathbb{R}^{C \times HW}$ represents the fused semantic-geometric features

of the current frame, as defined in Eq. 6. The perturbation weights $\mathbf{\Delta}'$ complement the semantic weights with geometric information and are defined as:

$$\mathbf{\Delta}' = P_{null}\Delta, \tag{10}$$

where $\Delta$ is obtained from Eq. 7 using the ***geometry model predictor***, and $P_{null} \in \mathbb{R}^{C \times C}$ is the null-space projection matrix computed from the semantic features.

Inspired by AlphaEdit, we use Singular Value Decomposition (SVD) to compute the null space projector $P_{null}$ for semantic features. Rank deficiency frequently arises in feature representations in the GOT setting, which leads to ill conditioning and must be handled carefully. To ensure stability prior to SVD, we first apply whitening (Kessy et al., 2018) to the semantic features to obtain normalized features $\mathbf{Z}$ and then compute the regularized correlation matrix $\mathbf{M}$:

$$\mathbf{M} = \mathbf{Z}\mathbf{Z}^\top + \lambda\mathbf{I}, \tag{11}$$

where $\lambda$ is a ridge regularization term (Hoerl & Kennard, 1970).

We then construct the raw projector $\hat{\mathbf{P}} = U_{null}U_{null}^\top$ by selecting the eigenvectors $U_{null}$ of $\mathbf{M}$ corresponding to low-energy eigenvalues (identifying the subspace with minimal semantic information). To mitigate numerical drift during online inference, we explicitly symmetrize (Ammari et al., 2012a;b) the projector:

$$P_{null} = \frac{1}{2}(\hat{\mathbf{P}} + \hat{\mathbf{P}}^\top). \tag{12}$$

This projector is then utilized in Eq. 10 to compute the geometry-aware perturbation weights.

Unlike AlphaEdit, which performs offline model editing by collecting all preserved knowledge as in Eq. 1, our GOT-Edit predicts both preserved weights and perturbation weights in an online manner, enabling adaptive integration of geometric knowledge into the semantic model.

**Box Regression.** A regression decoder $RegDec$ takes the semantic–geometry enriched classification score map and the current frame features as input to predict a regression score map that provides the target bounding box in image resolution:

$$d = \textbf{\textit{RegDec}}\left(p \cdot z_{cur}\right), \tag{13}$$

where the operator $\cdot$ denotes channel-wise broadcasting multiplication, and the regression decoder $RegDec$, as used in (Mayer et al., 2022; Chen et al., 2025a), employs four convolutional layers to produce four feature maps $\mathbf{d}$ in the *ltrb* (left, top, right, bottom) bounding box representation (Tian et al., 2019). The coordinates with the highest classification score in $\mathbf{p}$ are mapped onto the regression score map $\mathbf{d}$ for final bounding box prediction.

**Objective Function.** The training objective is identical to that of previous work (Mayer et al., 2022; Bhat et al., 2019), i.e.,

$$\mathcal{L} = \lambda_{cls}L_{cls}(\hat{p}, p) + \lambda_{giou}L_{giou}(\hat{d}, d), \tag{14}$$

where $\hat{p}$ and $\hat{d}$ are the ground-truth labels. The target classification loss $L_{cls}$ is a compound hinge loss as described in (Bhat et al., 2019), while the GIoU loss (Rezatofighi et al., 2019) $L_{giou}$ is used to supervise bounding box regression. $\lambda_{cls}$ and $\lambda_{giou}$ are scalar weights that control the contribution of each loss, and these hyperparameters are identical to those in (Mayer et al., 2022).

## 4 EXPERIMENTAL RESULTS

### 4.1 EXPERIMENTAL SETTING

Modern trackers are trained on large-scale datasets comprising tens of millions of training samples and millions of test samples. We detail this as follows.

**Training Data.** Like most trackers (Mayer et al., 2022; Chen et al., 2025a; 2023; Lin et al., 2024), we use the training splits of LaSOT, GOT10k, TrackingNet, and COCO for model training. Some

trackers (Kang et al., 2025; Liang et al., 2025) include VastTrack (Peng et al., 2024) for training; we provide a variant of our tracker trained with this new dataset. The training data for GOT-Edit rigorously follows the VOT2022 (Kristan et al., 2022) challenge protocol and the GOT-10K guidelines.

**Test Data.** We use the following datasets for tracker performance evaluation:

- **AVisT** (Noman et al., 2022): Designed for testing without a training set, it encompasses 120 short and long sequences, each averaging 664 frames under adverse visibility conditions.
- **NfS** (Galoogahi et al., 2017) and **OTB** (Wu et al., 2015): Used for testing without a training set, each dataset contains 100 sequences, with an average of 534 frames per sequence.
- **GOT-10k** (Huang et al., 2019): It has 420 short sequences with an average of 149 frames per sequence, featuring non-overlapping object classes in the training and test sets.
- **LaSOT** (Fan et al., 2019) and **TrackingNet** (Fan et al., 2019): They provide training data where test classes fully overlap with training classes. LaSOT has 280 long sequences with an average of 2k frames per sequence, and TrackingNet offers 511 short sequences, averaging 471 frames each.
- **VOT2020 (Kristan et al., 2020) and VOT2022 (Kristan et al., 2022)**: These are the 2020 and 2022 editions of the Visual Object Tracking challenge (VOT-ST2020 and VOT-STb2022).

**Evaluation Metrics.**

We evaluate trackers using the following metrics:

- **SUC** (success rate): The percentage of frames in which the predicted bounding box overlaps the ground truth by at least an IoU threshold or the average of all thresholds.
- **SR75**: It refers to SUC with an IoU threshold of $75\%$.
- **OP50**: The percentage of frames where the predicted and ground truth IoU exceed 50%.
- **Pr** (precision): It measures the percentage of frames where the predicted target center is within $T$ pixels of the ground-truth center. $T$ is set to 20 in this work.
- **NPr** (normalized precision): It is the percentage of frames where the center location error, normalized by the target's box diagonal, is less than the threshold of $0.2$.
- **AO** (average overlap): The mean IoU between the predicted and ground-truth bounding boxes.

**Implementation Details.** Our method is implemented using PyTorch 2.0.0 and CUDA 11.7. We train the model on eight NVIDIA RTX 4090 GPUs (24 GB each). DeepSpeed (Rasley et al., 2020) is integrated to accelerate training. We also verify that applying activation checkpointing to the tracker further reduces memory consumption, enabling training of the tracker at high resolution ($378 \times 378$) on four 24 GB GPUs. Inference is performed on a single NVIDIA RTX 4090 GPU and consumes approximately 9 GB of GPU memory during evaluation.

Following PiVOT (Chen et al., 2025a) and LoRAT (Lin et al., 2024), we use the DINOv2 (Oquab et al., 2023) ViT-L backbone for image feature extraction. We initialize the model predictors and localization head with pretrained ToMP-L weights, a ToMP variant with a DINOv2-L backbone. The backbone remains frozen during tracker training. For integrating geometric information, we extract intermediate features from the DPT head of VGGT (Wang et al., 2025a), which is kept frozen during training. We also evaluate an alternative geometry backbone using Depth Anything 3 (Lin et al., 2026), as in Table 1, and StreamVGGT (Zhuo et al., 2026), as in the Appendix.

For an efficient design, the dual model predictors share the same architecture and weights, but two independent lightweight convolutional layers are appended in parallel to the predictors, serving as task-specific heads for semantic weight prediction and perturbation weight prediction, respectively.

We sample 200K subsequences per epoch and train for 25 epochs. Each subsequence consists of two reference frames and one current frame, randomly selected from a 200-frame window within a video sequence. The frames to VGGT are concatenated spatially, which allows better geometric features through multi-frame interaction. Following ToMP (Mayer et al., 2022), PiVOT (Chen et al., 2025a), we set the search area scale factor to $5.0$ and perform data augmentation. The initial learning rate is set to $10^{-4}$ with a StepLR scheduler that decays it by a factor of 0.2 at epochs 10, 15, and 20. AdamW (Loshchilov & Hutter, 2019) is used as the optimizer.

To mitigate the computational cost of higher image resolutions, as in recent works (Xie et al., 2025; Li et al., 2025; Chen et al., 2023; Lin et al., 2024), we use smaller resolutions for most ablations

Table 1: **Comparison with state-of-the-art methods.** Each tracker is followed by its input resolution. The term 'Base' in the column 'Training Data of Tracker' refers to trackers trained on the classical four datasets. 'Frames' denotes the number of frames a tracker uses on each frame during evaluation. '*' denotes a tracker trained solely on the specific GOT-10k set (Huang et al., 2019).

| | | | | | | Low or No Overlap | | | | | Full Overlap | | | | |
| | | | Training-Test Class Overlap | | | AVisT | NfS | OTB | GOT-10k* | | LaSOT | | | TrackingNet | |
| | | | Dataset | | | | | | | | | | | | |
| Tracker | Semantic Feature | Geometry Feature | Training Data of Tracker | Frames | Trainable Parameters | SUC | SUC | SUC | AO | SR75 | NPr | Pr | SUC | NPr | SUC |
|---|---|---|---|---|---|---|---|---|---|---|---|---|---|---|---|
| **GOT-Edit-378 (Ours)** | DINOv2-L | VGGT | Base+VastTrack | 3 | 53M | 64.5 | 71.1 | 75.0 | 80.2 | 79.8 | 84.8 | 82.9 | 75.0 | 91.0 | 86.7 |
| **GOT-Edit-378 (Ours)** | DINOv2-L | VGGT | Base | 3 | 53M | 63.7 | 69.9 | 73.0 | | | 85.2 | 83.2 | 75.3 | 90.6 | 86.4 |
| **GOT-Edit-378 (Ours)** | DINOv2-L | DA3-L | Base+VastTrack | 3 | 28M | 64.7 | 70.8 | 74.8 | 79.5 | 79.6 | 85.2 | 83.4 | 75.5 | 91.0 | 86.6 |
| **GOT-Edit-252 (Ours)** | DINOv2-L | DA3-L | Base+VastTrack | 3 | 51M | 63.0 | 70.7 | 72.6 | 76.4 | 75.8 | 84.5 | 81.7 | 73.9 | 90.1 | 85.6 |
| PiVOT-378 (Chen et al., 2025a) | DINOv2-L | - | Base | 3 | 34M | 62.2 | 68.2 | 71.2 | 76.9 | 75.5 | 84.7 | 82.1 | 73.4 | 90.0 | 85.3 |
| LoRAT-378 (Lin et al., 2024) | DINOv2-L | - | Base | 3 | 32M | 62.0 | 66.7 | 72.0 | 77.5 | 78.1 | 84.1 | 82.0 | 75.1 | 89.7 | 85.6 |
| ToMP-378 (Chen et al., 2025a) | DINOv2-L | - | Base | 3 | 25M | 61.5 | 67.8 | 71.0 | - | - | 83.6 | 80.8 | 72.6 | - | - |
| ToMP-378 (Reproduced) | DINOv2-L | - | Base+VastTrack | 3 | 25M | 62.0 | 69.0 | 71.5 | 77.5 | 75.8 | 83.7 | 80.8 | 72.7 | 89.0 | 84.2 |
| MCITrack-384 (Kang et al., 2025) | Fast-iTPN-L | - | Base+VastTrack | 5 | 287M | 62.9 | 70.6 | 72.0 | 80.0 | 80.2 | 86.1 | 85.0 | 76.6 | 92.1 | 87.9 |
| ARPTrack-384 (Liang et al., 2025) | ViT-ARP-L | - | Base+VastTrack+K700 | 7 | 405M | - | - | - | 81.5 | 80.5 | 83.4 | 81.7 | 74.2 | 91.1 | 86.6 |
| SeqTrack-384 (Chen et al., 2023) | ViT-MAE-L | - | Base | 3 | 309M | 57.8 | 66.7 | - | 74.8 | 72.2 | 81.5 | 79.3 | 72.5 | 89.8 | 85.5 |
| GRM-320 (Gao et al., 2023) | ViT-MAE-L | - | Base | 3 | - | 54.5 | 66.9 | 68.9 | 73.4 | 70.4 | 81.2 | 77.9 | 71.4 | 88.9 | 84.0 |
| SATrack-384 (Ma et al., 2025) | SAViT | - | Base | 6 | - | 58.4 | 67.5 | - | 75.4 | 73.5 | 81.4 | 78.4 | 72.0 | 89.0 | 84.7 |
| DeTrack-384 (Zhou et al., 2024) | Denoising ViT | - | Base | 3 | - | 60.2 | - | - | 77.9 | 74.9 | 81.7 | 79.1 | 72.9 | - | - |

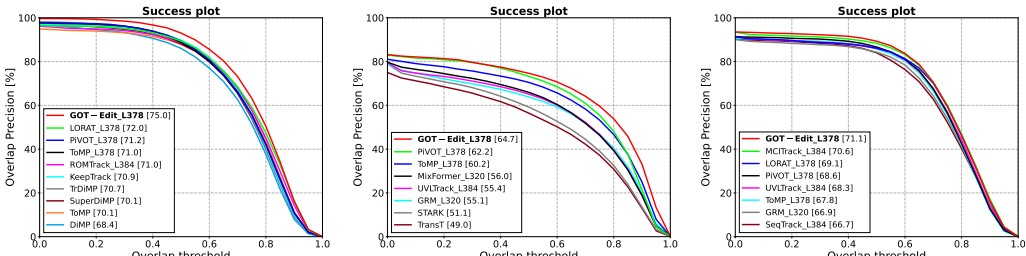

Figure 2: From left to right, success plots of competing methods on OTB, AVisT, and NfS are shown.

and higher resolutions for comparison with the state of the art: 1) **GOT-Edit-252**, where the frame resolution and the patch token size are $252 \times 252$ and $18 \times 18$, respectively; 2) **GOT-Edit-378**, where the frame resolution and the token size are $378 \times 378$ and $27 \times 27$, respectively. We also employ mixed-precision training with BFloat16 and Float32 (or TFloat32) for efficiency.

## 4.2 COMPARISONS WITH THE STATE-OF-THE-ART METHODS

Table 1 compares our GOT-Edit with the SOTAs on several benchmark datasets. When compared with trackers that use semantic backbones based on DINOv2 (Oquab et al., 2023), our tracker demonstrates superior performance, generalizes well to out-of-distribution targets, and achieves competitive results on in-distribution targets. GOT-Edit shows a performance gain of about 2–3% across datasets compared with ToMP-378, which is a DINOv2 variant of ToMP (Mayer et al., 2022) and serves as the baseline tracker. Comparing against trackers that employ different semantic backbones, our tracker outperforms all trackers on out-of-distribution targets, except MCITrack-384 (Kang et al., 2025) on in-distribution targets, which uses a different semantic backbone and involves more trainable parameters and frames during training and evaluation. In addition to SUC, NPr, and Pr, we compare trackers using OP50 (Table 3), where all trackers share the same semantic backbone, and GOT-Edit uses VGGT as the geometry backbone. Our tracker outperforms others by a clear margin on this metric. We also provide the success AUC curves in Figure 2, reporting the best-performing variants of our tracker alongside those of the compared methods. On OTB, our method consistently shows the best results. Our tracker outperforms all trackers when $T > 0.2$ on AVisT, while outperforming MCITrack when $T < 0.7$ on NfS. Additionally, we provide an evaluation on the VOT challenge in Table 2.

## 4.3 ABLATION STUDIES

Table 4 presents ablation studies on each GOT-Edit component under image resolution 252, trained using four classical datasets. Row (1) shows the baseline method trained using only semantic features. Row (2) shows that the GOT tracker takes features from the DPT head of VGGT. Even though these features are used to finetune the tracker with GOT data, performance still drops dramatically due to the limited discriminative ability of the geometric information. Row (3) shows the fusion of semantic features from VGGT's DINO head and geometric features from VGGT's DPT head, which yields a moderate improvement compared with using geometric features alone. Row

Table 2: Comparisons among trackers on the VOT challenge using Robustness as the metric.

| | GOT-Edit | PiVOT | MixFormerL | OSTrackSTB | TransT_M | ToMP |
|---|---|---|---|---|---|---|
| VOT-STb2022 | 89.8 | 87.3 | 85.9 | 86.7 | 84.9 | 81.8 |
| VOT-ST2020 | 90.3 | – | 85.5 | – | – | 78.9 |

Table 3: Comparison with trackers using DINO features under OP50.

| Dataset | AVisT | NfS | LaSOT |
|---|---|---|---|
| Tracker / Metric | | OP50 | |
| GOT-Edit-378_Vast | 74.4 | 89.3 | 85.9 |
| GOT-Edit-378 | 73.7 | 88.7 | 86.1 |
| ToMP-378 | 72.6 | 85.7 | 84.8 |
| LoRAT-378 | 72.4 | 85.6 | 85.1 |

Table 4: Ablation studies on GOT-Edit with several design choices compared across multiple datasets under SUC.

| | Semantic (DINO) | Semantic (VGGT's DINO) | Geometry (VGGT) | Null Space Constrain | Regulari-zation | AVisT | NfS | LaSOT |
|---|---|---|---|---|---|---|---|---|
| (1) | ✓ | | | | | 59.2 | 68.5 | 70.7 |
| (2) | | | ✓ | | | 55.8 | 66.3 | 67.6 |
| (3) | | ✓ | ✓ | | | 59.9 | 67.5 | 70.9 |
| (4) | ✓ | | ✓ | | | 60.2 | 68.5 | 71.3 |
| (5) | ✓ | | ✓ | ✓ | | 61.5 | 69.3 | 72.7 |
| (6) | ✓ | | ✓ | ✓ | ✓ | 62.0 | 70.2 | 73.8 |

Figure 3: Attribute analysis of OTB, AVisT, and LaSOT from left to right, with average scores below.

(4) shows semantic features extracted from an independent DINO backbone, which perform better than semantic features from the DINO head of VGGT. This effect can be largely attributed to the fine-tuning of the DINO backbone of VGGT with large-scale 3D data, which distorts the original semantic representations of the DINO backbone. Row (5) shows semantic–geometry fusion under the null-space constraint, which improves performance compared with fusion without the constraint. Row (6) shows that whitening and regularization applied to input features before SVD further improve overall performance.

Overall, our online model editing strategy for geometry–semantics combination improves the baseline by an average of $2.5\%$, while the null space constraint with regularization effectively enhances fusion, yielding notable gains across datasets: $1.8\%$ on AVisT, $1.7\%$ on NfS, and $2.5\%$ on LaSOT. These results demonstrate the superiority of GOT-Edit.

Our method freezes semantic and geometry feature extractors and fuses them using the proposed knowledge-editing approach during training, enabling seamless cooperation between the two modalities and further complements the semantic distortion in VGGT, where semantic features tend to be dominated by geometry, and complements existing GOT trackers, which lack geometric knowledge.

Table 5 shows the ablation studies of GOT-Edit-252 components with regard to attributes. Row (1) presents the baseline performance. Row (2) reports the results of incorporating semantic and geometric information under a naive fusion method. For attributes related to 3D (e.g., occlusion, visibility, background clutter), the performance improves. However, for non-3D-related attributes (e.g., distractor, fast motion, illumination), the performance degrades. By addressing the fusion balancing problem through the null-space constraint, as adopted in our GOT-Edit, the tracker achieves not only geometric benefits but also semantic consistency, as demonstrated in row (3).

### 4.4 COMPARISON OF ATTRIBUTES AMONG SOTA

We conduct an attribute-based analysis by comparing our GOT-Edit with several trackers like (Song et al., 2022; Zheng et al., 2024; Ma et al., 2024; Cui et al., 2022; Wang et al., 2021) using large resolution input, as shown in Figure 3. This analysis provides insights into the strengths and weaknesses of different methods and highlights potential areas for improvement. Note that attribute-based plotting requires the raw results of a tracker. If the raw data of a tracker is unavailable or if datasets lack an attribute analysis protocol (e.g., those hosted on third-party servers without attribute results), we exclude those trackers from the attribute analysis.

Table 5: Ablation studies of GOT-Edit components with regard to the attributes.

| | Semantic (DINO) | Geometry (VGGT) | Null Space Constrain | AVisT | | | | LaSOT | | | | |
| | | | | Weather Conditions (Target Visibility) | Obstruction Effects (Occlusion) | Camouflage (Background Clutter) | Target Effects (Distractor) | Partial Occlusion | Full Occlusion | Background Clutter | Fast Motion | Illumination |
|---|---|---|---|---|---|---|---|---|---|---|---|---|
| (1) | ✓ | | | 64.32 | 57.14 | 42.21 | 49.38 | 68.97 | 62.93 | 64.25 | 60.39 | 72.02 |
| (2) | ✓ | ✓ | | 66.58 | 59.83 | 44.37 | 47.18 | 70.08 | 63.74 | 65.45 | 58.73 | 71.13 |
| (3) | ✓ | ✓ | ✓ | 67.95 | 62.67 | 46.93 | 50.27 | 71.60 | 66.33 | 67.85 | 62.90 | 73.23 |

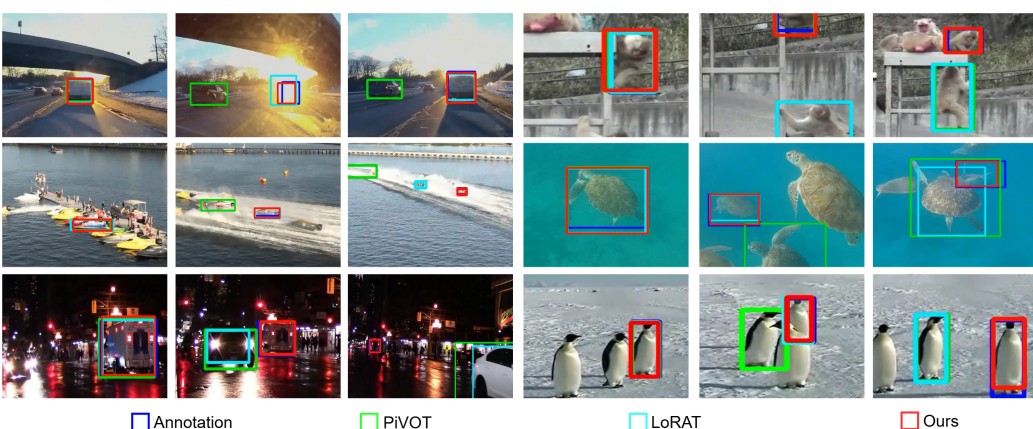

☐ Annotation     ☐ PiVOT     ☐ LoRAT     ☐ Ours

Figure 4: Visual comparisons of tracking results from GOT-Edit, PiVOT, and LoRAT across diverse video sequences under adverse scenarios are shown. The three left columns illustrate object tracking evaluation on AVisT, while the three right columns present tracking results on LaSOT.

**OTB**: As the left column of Figure 3 illustrates, our tracker achieves a considerable performance gain on attributes such as background clutter, occlusion, and rotation, compared with the baseline ToMP-L378. These improvements result from the geometry information that aids the understanding of the scene and the object itself, while other attributes still outperform competing trackers.

**AVisT**: As shown in the middle column of Figure 3, our tracker improves most attributes over other trackers. Although it trails PiVOT in Imaging Effects under low-light conditions, it still outperforms the baseline ToMP-L378 across attributes, demonstrating effectiveness on unseen data.

**LaSOT**: The right column of Figure 3 demonstrates that our tracker outperforms most attributes compared with other trackers; however, in viewpoint change and fast motion, it performs similarly or slightly drops below some trackers. This is because visual geometry becomes less effective when the scene or object moves rapidly or undergoes significant viewpoint changes.

**Limitations.** While improved in most attributes, our tracker still requires enhancement in handling moving objects, as evidenced in the LaSOT benchmark. The 'Target Effects' attribute in the AVisT benchmark, which contains distractors and fast-moving objects, provides evidence for improvement. Additionally, handling out-of-distribution data, as in AVisT, presents opportunities.

### 4.5 VISUALIZATION

We present visual comparisons among trackers in Figure 4. Our tracker is more robust to occlusion and better discriminates distractors, enabled by semantic and geometric reasoning.

## 5 CONCLUSION

We present GOT-Edit, the first framework to integrate 3D geometric cues into generic object tracking via online model editing, while using only 2D streaming inputs during tracking. By constraining updates to preserve semantics, GOT-Edit prevents semantic degradation while incorporating geometric cues that conventional 2D trackers overlook. Through online model editing with null-space constraint, it retains semantic knowledge while adaptively integrating geometric information, achieving robustness under occlusion, clutter, and visual ambiguity. The framework generalizes across datasets, targets, and environments while maintaining stability and robustness. Beyond surpassing state-of-the-art trackers in generalization, the results demonstrate that principled model editing can bridge modality gaps and recover geometry information missed by purely 2D approaches. These advances chart a path toward reliability, safety, and social responsibility in vision systems.

ETHICS STATEMENT

The proposed GOT-Edit framework improves generic object tracking by adaptively integrating semantic and geometric reasoning through online model editing. This capability offers potential societal benefits, including greater reliability of autonomous and robotic systems and improved assistance in challenging visual environments. However, the method may be misused for intrusive surveillance or other applications that compromise privacy and security. Deployment must therefore comply with legal and ethical standards, particularly in contexts involving personal data or sensitive environments. The tracker is trained solely on publicly available datasets, consistent with existing methods and in accordance with established ethical standards. As a pioneering method that enhances tracking by integrating visual geometry, GOT-Edit highlights the emerging, potentially imperfect nature of 3D inference from 2D inputs as a key direction for ongoing technical improvement. Responsible use requires transparency, rigorous validation, and adherence to established ethical guidelines.

REPRODUCIBILITY

To ensure reproducibility, detailed implementation instructions for GOT-Edit are provided in 4.1. The source code is publicly available at `https://github.com/chenshihfang/GOT`. These measures are intended to facilitate the verification and replication of the results by other researchers.

ACKNOWLEDGEMENT

This work was supported in part by the National Science and Technology Council (NSTC) under grants 112-2221-E-A49-090-MY3, 114-2221-E-A49-038-MY3, and 114-2634-F-002-004-, by Academia Sinica under grant AS-IAIA-114-M10 and by MediaTek. This work was supported by H100 GPU computing resources donated by Wistron.

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

## A  APPENDIX

This document supplements the main paper with details on GOT-Edit, including comparisons with state-of-the-art trackers using NPr, Pr, and SUC plots, ablation studies on model complexity, and a video appendix for qualitative visualization, available on the paper submission forum.

## B  COMPUTATIONAL COST ANALYSIS

Table 6: The analysis quantifies the computational costs of each component of the GOT-Edit in terms of runtime per frame (milliseconds, ms).

| Frame Resolution | Backbone | | Align and Fuse | Model Predictors | Reg/Cls Decoders | Total |
|---|---|---|---|---|---|---|
| | VGGT | DINO | | | | |
| $252 \times 252$ | 65.6 | 8.7 | 2.3 | 6.8 | 0.7 | 84.1 |
| $378 \times 378$ | 91.9 | 17.6 | 2.7 | 14.5 | 0.7 | 127.4 |

The computational cost of each tracker component is reported in Table 6 as per-frame runtime (ms). The primary computational overhead is dominated by geometric feature extraction (VGGT). Our core contribution, the online model editing modules (Align and Fuse and Model Predictors), is highly efficient, with a runtime of only 9.1 ms at a $252 \times 252$ frame resolution or 17.2 ms at a $378 \times 378$ resolution. The evaluation model uses BFloat16 for VGGT.

Table 7: Runtime and FLOPs breakdown for VGGT, DINO, and the tracker component.

| Frame Resolution | Metric | VGGT | DINO | Tracker Excluding VGGT & DINO |
|---|---|---|---|---|
| $252 \times 252$ | Runtime (ms) | 65.6 | 8.7 | 9.8 |
| | FLOPs (G) | 1000 | 105 | 32 |
| $378 \times 378$ | Runtime (ms) | 91.9 | 17.6 | 17.9 |
| | FLOPs (G) | 2253 | 251 | 73 |

We also provide the model complexity in terms of FLOPs (Floating-Point Operations), as shown in Table 7. FLOPs are agnostic to device and precision, and we compute MACs (multiply–accumulate operations and multiply) and multiply the result by two to obtain FLOPs.

## C  MORE EXPERIMENTS

**Analysis of Alternate Geometry Backbone Choices**

To enhance speed performance, we utilize StreamVGGT (Zhuo et al., 2026) to replace VGGT for geometric feature extraction and report the results in Table 8. In this table, 'GlobalAttn FineTune' refers to using DoRA (Liu et al., 2024b) to fine-tune the linear layers of the global attention layer in the geometry model, where the global attention layer is the key mechanism for handling cross-frame information. 'MemCache' refers to the number of historical K/V caches used for tracking. 'Frequency' denotes the frequency for geometric feature extraction. The DoRA rank is set to 16, and only 2.4 M parameters are fine-tuned for the geometry model. The experimental results in the table demonstrate that optimized geometric variants and selective feature application (we set the memory cache to 3 and apply geometric information every 3 frames in the StreamVGGT variant) can significantly increase the speed (e.g., runtime is reduced by approximately 40% when StreamVGGT replaces VGGT, while competitive accuracy is maintained.

**Analysis of Attribute-Wise Performance under Semantic and Geometry Configurations**

To explicitly evaluate the influence of both the geometric and semantic backbones, we conduct additional experiments 9 at the consistent resolution of $378 \times 378$. These extended experiments validate our method by varying both the semantic backbone (DINOv2 vs. MAE (He et al., 2022)) and the geometric backbone (VGGT vs. StreamVGGT). Experiments (1) and (3) in Table 9 establish the baselines using only the semantic backbones MAE-L and DINOv2-L, respectively. Once additional geometric backbones, VGGT and StreamVGGT, are adopted, our GOT-Edit can leverage the geometric features and substantially improve performance across various challenging attributes, such as occlusion, background clutter, and distractors.

Table 8: Efficiency in runtime (ms per frame) and accuracy (%) for VGGT and StreamVGGT with varying cache and update frequency.

| Tracker | Geometry Method | GlobalAttn FineTune | Mem Cache | Frequency | Runtime | LaSOT | AVisT | NfS |
|---|---|---|---|---|---|---|---|---|
| | VGGT | - | - | Every Frame | 84.1 | 73.8 | 62.0 | 70.2 |
| | | - | 1 | Every Frame | 72.5 | 72.8 | 61.4 | 69.7 |
| | | | 1 | Every Frame | 72.9 | 73.5 | 61.6 | 70.0 |
| GOT-Edit-252 | StreamVGGT | | 2 | Every 2 Frames | 59.4 | 72.3 | 61.8 | 69.5 |
| | | ✓ | 2 | Every 3 Frames | 53.9 | 72.7 | 62.0 | 69.2 |
| | | | 3 | Every 2 Frames | 67.8 | 73.1 | 62.7 | 70.0 |
| | | | 3 | Every 3 Frames | 56.2 | 73.4 | 61.9 | 69.8 |
| | VGGT | - | - | Every Frame | 127.4 | 75.0 | 64.5 | 71.1 |
| | | - | 2 | Every 2 Frames | 84.6 | 74.3 | 63.2 | 69.5 |
| GOT-Edit-378 | StreamVGGT | | 2 | Every 2 Frames | 84.0 | 74.9 | 64.1 | 70.9 |
| | | | 2 | Every 3 Frames | 72.4 | 74.8 | 64.3 | 70.7 |
| | | ✓ | 3 | Every 2 Frames | 92.1 | 74.8 | 63.2 | 71.2 |
| | | | 3 | Every 3 Frames | 78.4 | 75.2 | 63.3 | 71.4 |

Table 9: Attribute-wise performance with different semantic and geometry configurations.

| | Semantic | | Geometry | | AVisT | | | |
|---|---|---|---|---|---|---|---|---|
| | DINO | MAE | VGGT | StreamVGGT | Weather Conditions (Target Visibility) | Obstruction Effects (Occlusion) | Camouflage (Background Clutter) | Target Effects (Distractor) |
| (1) | | ✓ | | | 65.07 | 56.69 | 62.07 | 44.58 |
| (2) | | ✓ | ✓ | | 65.81 | 60.10 | 66.21 | 45.93 |
| (3) | ✓ | | | | 65.31 | 58.89 | 66.94 | 45.79 |
| (4) | ✓ | | | ✓ | 68.54 | 61.41 | 68.33 | 48.86 |
| (5) | ✓ | | ✓ | | 68.39 | 61.31 | 68.73 | 49.68 |

**NPr, Pr, and Suc Plots**

We report NPr, Pr, and SUC plots on four datasets: NfS, AVisT, LaSOT, and OTB. Other datasets, such as TrackingNet and GOT-10K, are evaluated on online servers without plots and thus excluded.

**Overview Guidelines for NPr, Pr, and Suc Plots:**

In the Precision (Pr) and Normalized Precision (NPr) plots, the x-axis denotes pixel or normalized distance thresholds, while the y-axis indicates the percentage of frames in which the distance between the predicted and ground-truth target centers falls within the specified threshold. A balance is typically sought between higher precision and lower localization error. Trackers are commonly ranked by their performance at a threshold of 0.2 in NPr or 20 pixels in Pr.

In the Success (SUC) plot, the x-axis represents the IoU thresholds (measuring the overlap between the predicted bounding box and the ground truth), while the y-axis indicates the percentage of frames in which the IoU meets or exceeds the corresponding threshold. Trackers are commonly ranked by their performance, measured as the average precision across all thresholds.

We analyze the plots for each dataset as follows:

- **NfS**: In Figure 5, our tracker outperforms others once the threshold exceeds 0.1 in NPr or 10 pixels in Pr. For SUC, it consistently surpasses all baselines across thresholds.

- **AVisT**: As shown in Figure 6, AVisT, a training-free dataset with diverse adverse scenarios. Under conditions NPr with $T < 0.3$, our tracker outperforms all baselines. For PR, our tracker outperforms competitors across thresholds. For SUC, our tracker outperforms competitors when $T > 0.4$.

- **OTB**: In Figure 7, our tracker consistently outperforms competitors e.g., (Lin et al., 2024; Chen et al., 2025a; Mayer et al., 2021; Wang et al., 2021) in SUC. For Pr and NPr, most trackers perform similarly, while our method remains significantly competitive.

- **LaSOT**: In Figure 8, on this in-distribution dataset, our tracker outperforms SOTA methods, e.g., (Zheng et al., 2024; Cai et al., 2023; Song et al., 2023; 2022) when NPr $T > 0.1$, Pr $T > 10$ pixels, and SUC $< 0.7$. LoRAT surpasses our tracker only under very strict conditions, such as NPr $T < 0.1$, Pr $T < 10$ pixels, and SUC $> 0.8$. Nevertheless, our method consistently outperforms other trackers with the same backbone, including PiVOT-L378 and ToMP-L378.

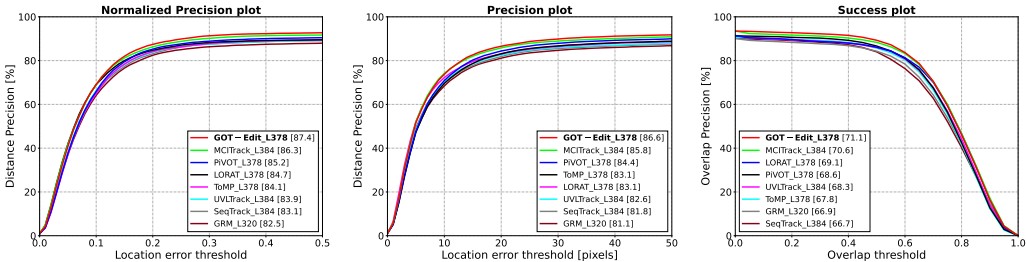

Figure 5: **Comparison of methods using NPr, Pr, and SUC on NfS, left to right.**

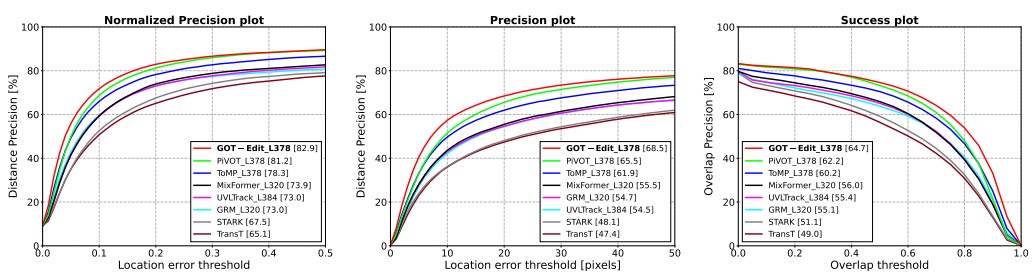

Figure 6: **Comparison of methods using NPr, Pr, and SUC on AVisT, left to right.**

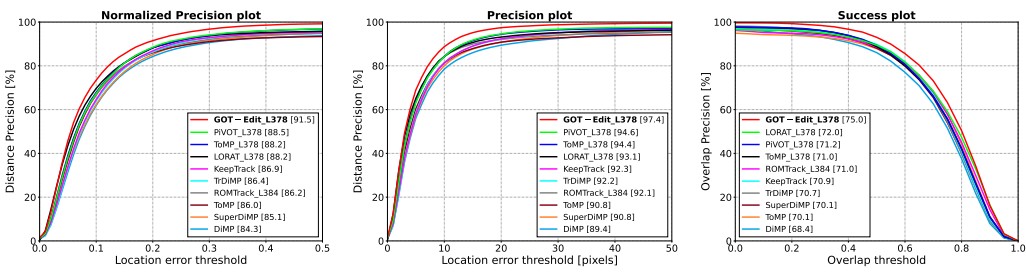

Figure 7: **Comparison of methods using NPr, Pr, and SUC on OTB, left to right.**

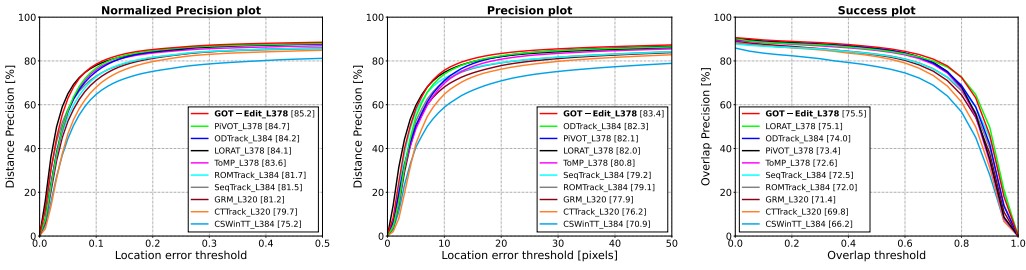

Figure 8: **Comparison of methods using NPr, Pr, and SUC on LaSOT, left to right.**

# D  THE USE OF LARGE LANGUAGE MODELS

The research is original, and large language models were used only for polishing the writing.

