# OpenReview forum: "GOT-Edit: Geometry-Aware Generic Object Tracking via Online Model Editing"
_ICLR.cc/2026/Conference — ICLR 2026 Poster_

### Official Review · Reviewer_ukrA · 2025-10-17

**Soundness:** 2
**Presentation:** 3
**Contribution:** 2
**Rating:** 2
**Confidence:** 4

**Summary:**

This paper proposes the GOT-Edit method to integrate visual geometric knowledge into general object trackers through online model editing. First, the Visual Geometric Grounded Transformer (VGGT) trained on large-scale 3D annotated data is used to directly extract reliable geometric cues from 2D images. Secondly, to achieve seamless fusion of geometry and semantics, an online model editing strategy with null space constraints is designed. This adaptively fuses geometric information while retaining the semantic knowledge learned by the tracker, making its performance better than the simple fusion of geometric and semantic cues in a variety of scenarios. Finally, through extensive experiments on multi-class GOT benchmark datasets, GOT-Edit performs well in robustness and accuracy, especially in occluded and cluttered scenes. It has established a new paradigm for combining 2D semantics and 3D geometric reasoning for the field of general object tracking.

**Strengths:**

**1. Problem identification aligns with practical needs:** This paper accurately identifies the performance bottlenecks of current 2D general object tracking (GOT) under conditions of occlusion, background interference, and drastic changes in target geometry/appearance. It also proposes a targeted solution by integrating 3D geometric information, addressing the core pain point of "lack of scene depth understanding" in tracking tasks.

**2. Technical adaptability is reasonable:** Addressing the limitations of existing 3D-assisted tracking methods, which rely on additional 3D inputs such as RGB-D and point clouds, this paper employs VGGT to extract geometric features from 2D images, addressing the unavailability of 3D data in real-world scenarios. Furthermore, this paper improves AlphaEdit's offline null space constraint to online model editing, meeting the real-time requirements of tracking dynamic targets and backgrounds.

**3. Comprehensive experimental design:** Performance is verified on multiple benchmark datasets (such as AVisT, NfS, LaSOT, and GOT-10k), covering scenarios such as "in-distribution/out-of-distribution objects" and "poor visibility." Detailed ablation experiments verify the effectiveness of components such as semantic/geometric feature sources, null space constraints, and input regularization, resulting in clear results (see Tables 4 and 5).

**Weaknesses:**

1. The paper's proposal to "use 3D geometric features to compensate for the shortcomings of 2D semantics" is not a new idea. Existing work (e.g., Tan et al., 2025a/b; Chen et al., 2025b) has attempted to use 3D information for tracking, but has relied on additional 3D input. This paper merely changes the "source of 3D input" to "extracted from 2D images" (using VGGT). This is essentially an "adjustment of the input method" rather than an "original breakthrough in the tracking mechanism," and does not propose a completely new geometric-semantic fusion logic.

2. Model Editing Module: This is directly based on the nullspace constraint of AlphaEdit (Fang et al., 2025), merely changing "offline editing" to "online updating." It lacks theoretical innovation (such as nullspace calculation methods or constraint optimization), making it an "engineering tweak" rather than an "algorithmic originality."

3. Overall Framework: This is entirely based on the "track-by-detection" paradigm of ToMP (Mayer et al., 2022), adding only two components: "VGGT geometric feature extraction" and "nullspace fusion." No new tracking architecture (such as feature interaction mechanisms or localization head design) has been designed. This essentially replaces and supplements components of the existing framework.

4. The paper claims that "nullspace constraints protect semantic features from degradation," but fails to explain why geometric features inevitably fall into the nullspace of semantic features.

5. Some existing work (such as Wang et al., 2025, which uses VGGT for point tracking) has attempted to extract geometric information from 2D images. The paper does not clearly explain the fundamental difference between GOT-Edit and such work in terms of "tracking task adaptation," emphasizing only that "point tracking lacks semantics" without demonstrating the uniqueness of its semantic-geometric fusion.

6. The paper only notes that it is "inadequate for scenes with rapid motion and viewpoint changes" without further analyzing the underlying causes (e.g., estimation errors of geometric features in dynamic scenes? Improper adaptation of the dynamic weights of semantic-geometric fusion?), nor does it propose potential solutions, demonstrating a lack of understanding of the technical bottlenecks.

**Questions:**

1. Compared to existing work that "extracts geometric information from 2D images to assist tracking without requiring additional 3D input" (e.g., tracking methods based on monocular depth estimation), what is the original difference in the core mechanism (non-input method) of GOT-Edit? If the sole contribution is "the introduction of VGGT + null-space constraints," why does this combination constitute an innovative contribution?

2. Besides "converting AlphaEdit from offline to online," are there any other original improvements (e.g., optimizing the calculation of the null-space projection matrix, dynamic constraint strength adjustment)? If the sole contribution is "online," please justify the necessity of these adjustments for the tracking task—why are the dynamic update mechanisms of existing online tracking methods (e.g., ToMP) ineffective?

3. Why does geometric information fail in LaSOT's "fast motion" and "viewpoint changes" scenarios? Is it because the geometric features extracted by VGGT are insufficiently accurate, or is the fusion mechanism not well-suited to dynamic scenes? Please provide additional quantitative analysis and ablation experiments (e.g., to separately verify the effectiveness of geometric features in dynamic scenes).

**Details Of Ethics Concerns:**

No Ethics Concerns

---

> ### Author Response · Authors · 2025-11-24
>
> We thank the reviewer for recognizing the importance of this work in addressing practical failure modes of 2D generic object tracking (GOT) under occlusion, clutter, and geometric variation. We also appreciate the acknowledgement that integrating 3D cues without extra sensors is a reasonable and adaptive design choice. The positive assessment of our experimental coverage and the systematic ablation studies (on semantic/geometric sources, null space constraints, and regularization) supports our goal of providing a practically motivated and systematically validated geometry-aware GOT framework. We address each of your concerns below and have revised the paper accordingly.
>
> ---
>
> > ### W1 \& Q1: The paper's proposal to use 3D geometric features to compensate for the shortcomings of 2D semantics is not a new idea. Existing work (e.g., Tan et al., 2025a/b; Chen et al., 2025b) has attempted to use 3D information for tracking, but has relied on additional 3D input. This paper merely changes the source of 3D input to be extracted from 2D images (using VGGT). This is essentially an adjustment of the input method rather than an original breakthrough in the tracking mechanism, and does not propose a completely new geometric-semantic fusion logic?
>
> We thank the reviewer for this comment and agree that leveraging 3D information to enhance 2D tracking is an existing concept. However, we respectfully assert that our contribution, GOT-Edit, is not merely an adjustment of the input method: It introduces a novel tracking mechanism for inferring 3D cues from a 2D video stream and asymmetrically fusing them with 2D features, making it distinct from existing works in the following two aspects:
>
>
> **No External 3D data:** The key difference between our GOT-Edit and the cited RGB-X trackers (FlexTrack [Tan et al., 2025a/b] and SUTrack [Chen et al., 2025b]) lies in the practical input requirement. Both FlexTrack and SUTrack require additional precomputed 3D data (e.g., depth maps), but the GOT task is typically performed on RGB video streams. GOT-Edit is, to our knowledge, the first object tracker that incorporates 3D geometric information for tracking while requiring only RGB input. This setting aligns with human vision, where observers can infer 3D geometry from a 2D plane.
>
> **Asymmetric, Online Geometric–Semantic Fusion:** Recognizing that 2D semantic features remain the dominant signal for tracking on video sequences, whereas 3D geometric features inferred from 2D video are complementary but less reliable, we observed that symmetric fusion with inferred 3D geometry can degrade semantic performance (row (1) vs. row (2) in Table 5 of the paper). To address this, GOT-Edit introduces an online model-editing mechanism with a null-space constraint. This design enables the tracker to incorporate complementary 3D geometric knowledge without degrading the dominant 2D semantic features.
>
> We appreciate this comment and have revised paragraph 2 of Section `3D Features for Tracking' in the related work to clarify this point more clearly.

---

> ### Author Response · Authors · 2025-11-24
>
> > ### W2 \& Q2: Model Editing Module: This is directly based on the nullspace constraint of AlphaEdit (Fang et al., 2025), merely changing offline editing to online updating?
>
> We acknowledge that our module is developed upon the null-space constraint introduced by AlphaEdit. However, adapting this concept from offline editing to the online editing paradigm required for generic object tracking is a crucial contribution. This adaptation makes our method distinct from AlphaEdit in the following key aspects:
>
> **Online Model Editing for GOT:**
> AlphaEdit requires the precomputation and storage of a large-scale representation of old knowledge to define the preserved knowledge space, which is impractical for online generic object tracking due to varying data distributions between training and testing. To address this limitation, GOT-Edit introduces an online mechanism that uses dynamically updated few-shot examples to jointly predict and edit semantic and geometric model weights (Eqs. 6, 7, 9, and 10 of the paper). The null-space projector is computed from semantic features and applied to project geometric model weights into this null space. This design enables asymmetric fusion of the two modalities, ensures geometry-aware updates with semantic preservation for the tracker, and effectively adapts the AlphaEdit concept to the online tracking setting.
>
> **Robust Numerical Regularization for Online SVD:**
> We calculate the null-space projector using singular value decomposition (SVD). SVD requires a well-conditioned input matrix, but rank deficiency frequently occurs in features in tracking, leading to ill conditioning. While AlphaEdit can filter out bad sampling in an offline manner, we need to incrementally handle this issue for online tracking purposes. Specifically, as we unfold in the Online Model Editing section of the Method of the paper, we apply per-channel whitening and ridge regularisation to the features, which stabilizes null-space selection. Furthermore, we symmetrize the null-space projector to ensure numerical stability when using mixed precision. These stages of regularization successfully alleviate performance drops due to instability.
>
> **Cross-Modality Fusion:**
> AlphaEdit was designed for knowledge modifications within a single modality, e.g., large language models. In contrast, GOT-Edit is specifically designed to edit 2D semantic knowledge and 3D geometric knowledge across modalities, and is the first cross-modality model editing framework in this setting.
>
> We have revised the paragraph' Online Model Editing' in the Methods section, including Equations 11 and 12 of the paper, to clarify this point.

---

> ### Author Response · Authors · 2025-11-24
>
> > ### W3 \& Q2: This is entirely based on the `track-by-detection' paradigm of ToMP (Mayer et al., 2022). No new tracking architecture (such as feature interaction mechanisms or localization head design) has been designed?
>
> We acknowledge that GOT-Edit adopts the conventional track-by-detection paradigm and the architecture of ToMP. This decision was made intentionally to isolate and highlight our core contribution, online model editing.
>
> The primary goal of this work is to infer 3D knowledge to enhance 2D semantic reasoning for object tracking. To this end, our GOT-Edit provides a capability absent in conventional track-by-detection frameworks: it incorporates geometric knowledge without compromising semantic stability, allowing the tracker to adapt itself using only standard video streams as input.
> By building upon an existing method, we ensure a controlled comparison that clearly demonstrates the benefits of our model editing module.
> This design also enables a plug-and-play mechanism for modality fusion, which can be easily incorporated into other similar tracking paradigms.

---

> ### Author Response · Authors · 2025-11-24
>
> > ### W4 \& Q2. The paper claims that `nullspace constraints protect semantic features from degradation,' but fails to explain why geometric features inevitably fall into the nullspace of semantic features.
>
> We want to clarify that we do NOT assume that geometric features naturally fall into the null space of semantic features. Instead, our method explicitly projects the geometric model weights into the null space defined by the semantic features, as detailed in the Online Model Editing section.
>
> As mentioned in the Online Model Editing subsection of the paper, we empirically observe that a naive fusion of geometric and semantic information degrades the tracker's semantic capability, even when the performance on geometry-related attributes improves (as shown in Table 5). To mitigate this harmful interference, our method predicts the weights for both the semantic and geometric tracking models online. We then compute a null-space projection matrix using semantic weights and apply it directly to the geometry weights. This ensures that only the component orthogonal to the semantic span modifies the tracker. By construction, this update prevents the degradation of semantic knowledge while allowing geometry to contribute along directions orthogonal to semantic variation, consistent with the theoretical preservation guarantee of null-space editing in AlphaEdit.

---

> ### Author Response · Authors · 2025-11-24
>
> > ### W5 \& Q3. Some existing work, such as Wang et al. (2025), which uses VGGT for point tracking, has attempted to extract geometric information from 2D images. The paper does not clearly explain the fundamental difference between GOT-Edit and such work in terms of tracking task adaptation, emphasizing only that point tracking lacks semantics without demonstrating the uniqueness of its semantic-geometric fusion?
>
> We find that using geometric features alone for generic object tracking results in reduced performance compared to using semantic features, as shown in Table 4 (Row 1 vs. Row 2) of the paper.
>
> While VGGT [Wang et al., 2025] demonstrates that learned 3D geometric representations are useful for tasks like point tracking, it does not consider the joint use of semantic and geometric information, as point tracking optimizes pixel correspondences and is agnostic to the semantic identity of the target.
> In contrast, generic object tracking requires strong semantic discrimination to distinguish the target from distractors in dynamic scenes, where these attributes cannot be addressed solely by using geometry.
>
> The proposed GOT-Edit differs fundamentally by employing a unique online model editing mechanism to adapt these geometric features for the GOT task. As shown in Table 4 (Rows 4 vs. 6), naive fusion yields limited gains. Our method specifically projects geometry into the null space of the semantic model. This results in a semantic–geometry-aware tracking model that benefits generic object tracking by enhancing robustness without compromising the dominant semantic information.
>
> We appreciate this comment and have revised paragraph 2 of Section `3D Features for Tracking' in the related work to clarify this point more clearly.

---

> ### Author Response · Authors · 2025-11-24
>
> > ### W6 \& Q3. The paper only notes that it is `inadequate for scenes with rapid motion and viewpoint changes' without further analyzing the underlying causes (e.g., estimation errors of geometric features in dynamic scenes? Improper adaptation of the dynamic weights of semantic-geometric fusion?), nor does it propose potential solutions, demonstrating a lack of understanding of the technical bottlenecks?
>
>
> We appreciate the insightful comment regarding the need for a more in-depth analysis of performance bottlenecks, particularly in scenes with rapid motion and frequent changes in viewpoint.
>
> Our analysis indicates that the moderate performance improvement observed in the Fast Motion and Viewpoint Change (View Change) attributes (as detailed in Figure 3 and Table 5 of the paper), as well as the additional experiments showing that use of only the geometric backbone performs worse than use of only the 2D backbone in fast motion and viewpoint change scenarios (row (1) vs. row (2) in Table A), is primarily due to the constraint of using the geometric backbone, VGGT, with parameters frozen in our setting. The publicly released VGGT model is optimized for general 3D vision tasks but is not explicitly fine-tuned on dynamic scenes (e.g., point tracking or object tracking data).
>
> Therefore, the geometric features provided by the frozen VGGT, while highly effective for recovering 3D cues for static, occluded, or cluttered objects, are sub-optimally adapted for dynamic changes in appearance and geometric shifts. The core technical bottleneck is the lack of domain adaptation in the geometric feature extractor for dynamic object tracking scenarios.
>
> To investigate this technical bottleneck and validate our hypothesis, we conducted a new set of experiments (row (3) vs. row (4) in Table A and in Table B) involving fine-tuning the geometric backbone and exploring different online adaptation strategies using a faster geometric model, StreamVGGT [1]. This supplementary analysis demonstrates our in-depth understanding of the underlying technical bottleneck.
>
> In these tables, `GlobalAttn FineTune' refers to using DoRA [2] to fine-tune the linear layers of the global attention layer in the geometry model, where the global attention layer is the key mechanism for handling cross-frame information. The DoRA rank is set to 16, and only 2.4 M parameters are fine-tuned.
>
> ---
> **Table A. Comparison of feature configurations for tracking in fast motion and view change.**
>
> |      | Semantic       | Geometry       | GlobalAttn FineTune | View Change | Fast Motion |
> |:----:|:--------------:|:--------------:|:-------------------:|:-----------:|:-----------:|
> | (1)  | $\checkmark$   |                |                     | 71.0        | 60.4        |
> | (2)  |                | $\checkmark$   |                     | 66.2        | 54.9        |
> | (3)  | $\checkmark$   | $\checkmark$   |                     | 73.8        | 61.4        |
> | (4)  | $\checkmark$   | $\checkmark$   | $\checkmark$        | 74.3        | 64.2        |
>
>
> ---
> ---
>
> **Table B. Effect of geometry and global attention fine-tuning on tracking performance.**
>
> | Tracker           | Geometry       | GlobalAttn FineTune | LaSOT | AVisT | NfS |
> |:------------------|:-------------:|:-------------------:|:-----:|:-----:|:---:|
> | Baseline (ToMP-L) | -             | -                   | 72.7  | 62.0  | 69.0 |
> | GOT-Edit          | $\checkmark$  | -                   | 74.3  | 63.2  | 69.5 |
> | GOT-Edit          | $\checkmark$  | $\checkmark$        | 74.9  | 64.1  | 70.9 |
> ---
>
> The optimal solution to address the limitations in dynamic scenes is therefore to use a geometric backbone pre-trained on large-scale dynamic scene data or to implement more comprehensive, task-specific fine-tuning of the geometric feature extractor. We have integrated this analysis and the supplementary table into the revised paper.
>
> **References**
>
> [1] D. Zhuo et al. Streaming4D Visual Geometry Transformer. arXiv:2507.11539, 2025.
>
> [2] S.-Y. Liu et al. DoRA: Weight-Decomposed Low-Rank Adaptation. In ICML, 2024.
>
>
> - - -
>
> We are grateful for the reviewer’s recognition of the strengths of this work as well as the suggestions for improvement. We will gladly clarify any unresolved concerns.

---

> > ### Comment · Reviewer_ukrA · 2025-11-27
> >
> > Thanks the authors for providing these detailed replies. Most of my concerns are addressed and I would like to upgrade to 6.

---

> > > ### Author Response · Authors · 2025-11-27
> > > **Thank you.**
> > >
> > > Dear Reviewer,
> > >
> > > We are encouraged by this positive update! We sincerely thank you for your constructive suggestions regarding the presentation; your advice greatly helped us refine the manuscript.
> > >
> > > Paper 432 Authors

---

### Official Review · Reviewer_HoKn · 2025-10-27

**Soundness:** 3
**Presentation:** 3
**Contribution:** 3
**Rating:** 6
**Confidence:** 4

**Summary:**

This paper proposed GOT, which, by introducing the geometric knowledge of VGGT, achieves highly competitive experimental results with smaller trainable parameters.

**Strengths:**

1.The motivation of this paper is quite clear, and the performance-related experiments have well demonstrated the contribution. It is reasonable that information from multiple views (2D + 3D) can be better represented.
2.Sufficient visualization clearly demonstrates the improvement of GOT compared to the baseline. The dataset involved in the experiment is sufficient to demonstrate the generalization ability of GOT.
3.The author's most significant contribution lies in demonstrating the existence of the missing geometric prior and the necessity of integration, and proposing an effective editing method to integrate multi-channel information.

**Weaknesses:**

1.The ablation experiment results in Table 5 seem unable to be compared horizontally with those in Table 1. There are also many models based on DiNOv2-L, and their performances vary. Therefore, the backbone part is still worth comparing.
2.I have concerns about parameter and inference efficiency. Specifically, Table 6 shows the proportion of reasoning time for each part. The VGGT introduced as an innovation point accounts for the vast majority, and no horizontal comparison is made. The author only presented the trainable parameters and did not show the complete cost of model deployment. In fact, the results of some methods in specific scenarios (e.g., Table 1, Full Overlap column, PiVOT-378 row) do not differ much from GOT. Stating these differences clearly helps to clarify contributions.

**Questions:**

1.Apart from the limitations in comparison with related work, what improvement directions does the author think there are for GOT itself (e.g., the utilization of backbone and the future work of GOT)?
2.Which of the author 's experiments completed the declaration of 'without degrading the dominant semantic features'(on line 100)?
3.The author believes that the most core contribution is the creation of the engineering model or the proposal of an editing method that can effectively integrate the information of the two types of models?

---

> ### Author Response · Authors · 2025-11-24
>
> We thank the reviewer for the positive comments on the clarity of the motivation and on the experimental evidence for the contribution. We appreciate the recognition that multi-view information in the form of 2D semantics plus 3D geometry can better represent targets, that the visualizations and benchmarks demonstrate generalization, and that the main contribution lies in identifying missing geometric prior and proposing an effective editing method for integration. We address each of your concerns below and have revised the paper accordingly.
>
> ---
>
>
> > ### W1. The ablation experiment results in Table 5 seem unable to be compared horizontally with those in Table 1. There are also many models based on DiNOv2-L, and their performances vary. Therefore, the backbone part is still worth comparing.
>
> We appreciate the review for clarification regarding our experimental design. We address the concerns regarding comparison and backbone dependency in the following sections, respectively.
>
> **Experimental objectives:** The purposes of the experiments in Table 5 and Table 1 are fundamentally distinct. Thus, their results cannot be horizontally compared.
> Table 1 serves as the global benchmark for comparing the overall performance of competing methods, reporting tracking accuracy (e.g., SUC and NPr) averaged across all frames in a sequence against state-of-the-art literature. Tables of this type are widely used in the related papers.
>
> Conversely, Table 5 presents an attribute-wise analysis of our method exclusively. It reports the attribute-wise performance (e.g., Occlusion and Camouflage) computed only on frames annotated with that specific attribute. This table is designed for within-table comparison to validate how specific components (e.g., the null-space constraint and geometry integration) affect the tracker's behavior under fixed adverse conditions.
>
> **Backbone Comparison:** To explicitly evaluate the influence of both the geometric and semantic backbones, as suggested, we conduct additional experiments at the consistent resolution of $378\times378$ (matching the resolution used in Table 1). These extended experiments validate our method by varying both the semantic backbone (DiNOv2 vs. MAE[1]) and the geometric backbone (VGGT vs. StreamVGGT).
>
> **Table A. Attribute-wise performance with different semantic and geometry configurations.**
>
> |      | Semantic DiNO | Semantic MAE | Geometry VGGT | Geometry StreamVGGT | Weather Conditions | Obstruction Effects | Camouflage | Target Effects |
> |:----:|:-------------:|:------------:|:--------------:|:--------------------:|:------------------:|:-------------------:|:-----------:|:--------------:|
> | (1)  |               | $\checkmark$             |                |                      | 65.07              | 56.69               | 62.07       | 44.58          |
> | (2)  |               | $\checkmark$             | $\checkmark$               |                      | 65.81              | 60.10               | 66.21       | 45.93          |
> | (3)  | $\checkmark$              |              |                |                      | 65.31              | 58.89               | 66.94       | 45.79          |
> | (4)  | $\checkmark$              |              |                | $\checkmark$                     | 68.54              | 61.41               | 68.33       | 48.86          |
> | (5)  | $\checkmark$              |              | $\checkmark$               |                      | 68.39              | 61.31               | 68.73       | 49.68          |
>
> Experiments (1) and (3) in Table A establish the baselines using only the semantic backbones MAE-L and DiNOv2-L, respectively. Once additional geometric backbones, VGGT and StreamVGGT, are adopted, our GOT-Edit can leverage the geometric features and substantially improve performance across various challenging attributes, such as occlusion, background clutter, and distractors.
>
> We included these additional experiments in the subsection *Analysis of Attribute-wise Performance under Semantic and Geometry Configurations* in the appendix (following the references section of the main paper).
>
> **References**
>
> [1] K. He et al. Masked Autoencoders Are Scalable Vision Learners. In CVPR, 2022.
>
> [2] D. Zhuo et al. Streaming 4D Visual Geometry Transformer. arXiv:2507.11539, 2025.

---

> ### Author Response · Authors · 2025-11-24
>
> > ### W2. I have concerns about parameter and inference efficiency. Specifically, Table 6 shows the proportion of reasoning time for each part. The VGGT introduced as an innovation point accounts for the vast majority, and no horizontal comparison is made. The author only presented the trainable parameters and did not show the complete cost of model deployment. In fact, the results of some methods in specific scenarios (e.g., Table 1, Full Overlap column, PiVOT-378 row) do not differ much from GOT. Stating these differences clearly helps to clarify contributions.
>
> We clarify that the primary computational overhead is dominated by geometric (VGGT) feature extraction, as shown in Table B. Our core contribution, the online model editing modules included in `Tracker Excluding VGGT & DINO Backbone`, is efficient, with a runtime of only 9.8 ms at a $252 \times 252$ frame resolution or 17.9 ms at a $378 \times 378$ resolution.
>
>
> As requested, we have included a detailed comparison of model complexity (FLOPs) of the core components in Table B and comparisons of runtime, parameters (both trainable and total), and accuracy in success AUC across the baselines in Table C. Note that the input sequence length is $3$ for VGGT, whereas it is $1$ for DINO, which is consistent with the inference setting used during tracking.
>
> **Table B. Runtime (ms per frame) and FLOPs breakdown for VGGT, DINO, and the tracker component at different resolutions.**
>
> | Frame Resolution | Metric       | VGGT | DINO | Tracker Excluding VGGT & DINO |
> |:----------------:|:------------:|:----:|:----:|:------------------------------:|
> | 252 × 252        | Runtime (ms) | 65.6 | 8.7  | 9.8                            |
> |                  | FLOPs (G)    | 1000 | 105  | 32                             |
> | 378 × 378        | Runtime (ms) | 91.9 | 17.6 | 17.9                           |
> |                  | FLOPs (G)    | 2253 | 251  | 73                             |
>
>
>
>
> ---
> ---
>
> **Table C. Comparison of latency, parameter Count, and benchmark performance.**
>
> | Tracker             | Geometry Runtime (ms) | Semantic Runtime (ms) | Tracker (w/o Backbone) | Trainable Params | Total Params | AVisT | OTB  | GOT-10K | LaSOT |
> |:-------------------:|:---------------------:|:----------------------:|:-----------------------:|:----------------:|:-------------:|:-----:|:----:|:--------:|:-----:|
> | ToMP (DINO)         | –                     | 17.6                   | 16.1                    | 25M              | 0.3B          | 61.5  | 71.0 | 77.5     | 72.6  |
> | PiVOT (DINO+CLIP)   | –                     | 96.9                   | 17.5                    | 34M              | 0.6B          | 62.2  | 71.2 | 76.9     | 73.4  |
> | GOT-Edit (DINO+VGGT)| 91.9                  | 17.6                   | 17.9                    | 53M              | 1.3B          | 63.7  | 73.0 | 80.2     | 75.3  |
>
> The complexity of integrating geometric information is justified by the resultant performance gains, as shown in Table C. All experiments in this table compare training data using the classical four datasets (excluding the VastTrack dataset) for fair comparison, and all trackers use DINO as the backbone.
>
> Compared with the baseline ToMP, GOT-Edit shows a clear margin over it across multiple benchmarks.
> GOT-Edit outperforms PiVOT on the out-of-distribution benchmarks, including AVisT (63.7 vs. 62.2), OTB (73.0 vs. 71.2), and GOT-10K (80.2 vs. 76.9). For full-overlap datasets such as LaSOT, GOT-Edit also outperforms PiVOT in Success AUC (75.3 vs. 73.4). PiVOT utilizes CLIP as an auxiliary feature (DINO+CLIP) for discriminating among distractors, whereas our tracker focuses on leveraging geometry and semantics (DINO+VGGT).
>
> Although the number of parameters in GOT-Edit is larger than that of trackers that rely solely on semantic features, this primarily stems from the VGGT backbone (1B parameters) and the DINO backbone (304M parameters). Crucially, **we do not fine-tune parameters of these backbones on tracking data; they remain frozen during tracker training**. We update only the lightweight tracker modules (53M parameters), which enables efficient training.
>
>
> To reduce latency, we implement a speed-up version of the geometry model (StreamVGGT), with detailed results reported in Table D (please refer to the response for Q1, Table D).

---

> ### Author Response · Authors · 2025-11-24
>
> > ### Q1. Apart from the limitations in comparison with related work, what improvement directions does the author think there are for GOT itself (e.g., the utilization of backbone and the future work of GOT)? (1/2)
>
> This is a thought-provoking question. For generic object tracking, one primary limitation lies in achieving human-level robustness; we believe future directions should focus on incorporating semantic and geometric knowledge for 2D tracking, a path GOT-Edit begins to explore. This approach moves beyond adaptability (as seen in DiMP/ToMP) or contrastive analysis (like PiVOT) and represents a promising, yet underexplored, avenue toward superior generalization and modality cooperation.
>
> For our proposed method, GOT-Edit, the utilization of the backbone can be improved through two main efforts: developing a better geometry backbone for dynamic scene understanding and ensuring more efficient geometry feature prediction. We have partially addressed these by adopting the faster variant of VGGT (StreamVGGT) to reduce latency and by fine-tuning the geometry backbone's global attention layer to improve its relevance for 2D tracking within dynamic scenes.
>
> To enhance speed performance, we utilize StreamVGGT [1] to replace VGGT for geometric feature extraction and report the results in Table D. In addition, in Table D, `GlobalAttn FineTune` refers to using DoRA [2] to fine-tune the linear layers of the global attention layer in the geometry model, where the global attention layer is the key mechanism for handling cross-frame information. `MemCache` refers to the number of historical K/V caches used for tracking. `Frequency` denotes the frequency for geometric feature extraction. The DoRA rank is set to 16, and only 2.4 M parameters are fine-tuned.
>
> ---
>
> **Table D.  Efficiency in runtime (ms per frame) and accuracy (%) for VGGT and StreamVGGT with varying cache and update frequency.**
>
> | Tracker       | Geometry Method | GlobalAttn FineTune | Mem Cache | Frequency      | Runtime | LaSOT | AVisT | NfS  |
> |:-------------:|:---------------:|:-------------------:|:---------:|:--------------:|:-------:|:-----:|:-----:|:----:|
> | GOT-Edit-252  | VGGT            | -                   | -         | Every Frame    | 84.1    | 73.8  | 62.0  | 70.2 |
> | GOT-Edit-252  | StreamVGGT      | -                   | 1         | Every Frame    | 72.5    | 72.8  | 61.4  | 69.7 |
> | GOT-Edit-252  | StreamVGGT      | ✓                   | 1         | Every Frame    | 72.9    | 73.5  | 61.6  | 70.0 |
> | GOT-Edit-252  | StreamVGGT      | ✓                   | 2         | Every 2 Frames | 59.4    | 72.3  | 61.8  | 69.5 |
> | GOT-Edit-252  | StreamVGGT      | ✓                   | 2         | Every 3 Frames | 53.9    | 72.7  | 62.0  | 69.2 |
> | GOT-Edit-252  | StreamVGGT      | ✓                   | 3         | Every 2 Frames | 67.8    | 73.1  | 62.7  | 70.0 |
> | GOT-Edit-252  | StreamVGGT      | ✓                   | 3         | Every 3 Frames | 56.2    | 73.4  | 61.9  | 69.8 |
> | —             | —               | —                   | —         | —              | —       | —     | —     | —    |
> | GOT-Edit-378  | VGGT            | -                   | -         | Every Frame    | 127.4   | 75.0  | 64.5  | 71.1 |
> | GOT-Edit-378  | StreamVGGT      | -                   | 2         | Every 2 Frames | 84.6    | 74.3  | 63.2  | 69.5 |
> | GOT-Edit-378  | StreamVGGT      | ✓                   | 2         | Every 2 Frames | 84.0    | 74.9  | 64.1  | 70.9 |
> | GOT-Edit-378  | StreamVGGT      | ✓                   | 2         | Every 3 Frames | 72.4    | 74.8  | 64.3  | 70.7 |
> | GOT-Edit-378  | StreamVGGT      | ✓                   | 3         | Every 2 Frames | 92.1    | 74.8  | 63.2  | 71.2 |
> | GOT-Edit-378  | StreamVGGT      | ✓                   | 3         | Every 3 Frames | 78.4    | 75.2  | 63.3  | 71.4 |
>
>
> ---
>
> [Response continued in the next post]

---

> ### Author Response · Authors · 2025-11-24
>
> > ### Q1. Apart from the limitations in comparison with related work, what improvement directions does the author think there are for GOT itself (e.g., the utilization of backbone and the future work of GOT)? (2/2)
>
> [This is the continued response to the previous post]
>
> One key observation from Table D is that the StreamVGGT variant configured with three cache memories (Mem Cache = 3) and a frequency of every three frames (e.g., resolution $252 \times 252$), or with two cache memories and a frequency of every two frames (e.g., resolution $378 \times 378$), performs comparably to the original VGGT variant while reducing runtime by approximately 40%.
> Geometry–semantic knitting for tracking in the GOT-Edit StreamVGGT variant consumes only 56.2 ms per frame (GOT-Edit-252, on an NVIDIA GeForce RTX 4090 with PyTorch 2.0.0 and CUDA 11.7).
>
>
> In addition, the computational cost is a necessary price for integrating geometric reasoning, which is justified by the superior accuracy (Table 4 of the paper) and robustness (Table 5 of the paper).
>
> We have updated the sections *Computational Cost Analysis* and *Analysis of Alternate Geometry Backbone Choices* in the appendix to reflect these results.
>
> [1] D. Zhuo et al. Streaming 4D Visual Geometry Transformer. arXiv:2507.11539, 2025.
> [2] S.-Y. Liu et al. DoRA: Weight-Decomposed Low-Rank Adaptation. In ICML, 2024.
>
>
>
>
> > ### Q2. Which of the author's experiments completed the declaration of “without degrading the dominant semantic features” (on line 100)?
>
> Our attribute table (Table 5) shows that naïve fusion sometimes harms semantic attributes, such as Distractor, Fast Motion, and Illumination, whereas our null-space-projected edits restore them while maintaining the geometry gains.

---

> ### Author Response · Authors · 2025-11-24
>
> > ### Q3. The author believes that the most core contribution is the creation of the engineering model or the proposal of an editing method that can effectively integrate the information of the two types of models?
>
>
> We thank the reviewer for this thoughtful question.
> The core contribution of this work is an online editing method that facilitates integration of semantic and geometric modalities in the tracker, while the engineering design is essential to instantiate and validate this method in a realistic generic object tracking setting.
>
> On the methodological side, the main contribution is an **online**, semantics-preserving model editing mechanism that fuses semantic and geometric tracking models under a null-space projection constraint, enhancing geometric cues while preserving semantic knowledge. This mechanism is specifically adapted to the online setting through feature standardisation and regularisation, which mitigate the rank deficiency of the feature correlation matrix that frequently arises in online tracking.
>
> On the engineering side, we integrate VGGT (and StreamVGGT) into a generic object tracking framework and implement cross-modality online model editing for practical deployment.
> This design reflects generic object tracking, where semantic features provide the dominant cue and geometric information offers complementary guidance that fills missing geometric details. Naive or symmetric fusion with inferred 3D geometric features can degrade semantic performance, and the proposed online null-space constrained model editing in GOT-Edit addresses this challenge by enforcing geometry-aware updates that respect semantic knowledge. The engineering integration demonstrates effectiveness of this mechanism in a realistic GOT benchmark.
>
> In summary, the engineering design instantiates and validates the proposed online editing mechanism in a realistic GOT setting, while the editing rule itself forms the core conceptual contribution of the work.
>
> - - -
>
> Thank you again for the insightful and constructive feedback, which helped refine the presentation and positioning of the contributions of this work.
> If there is anything you would like to discuss further, we will be glad to respond.

---

> > ### Comment · Reviewer_HoKn · 2025-11-27
> > **Officia Comment by Reviewer HoKn**
> >
> > Thanks for the author's reply. The author has addressed most of the questions I was concerned about. Overall, I tend to  maintain my initial score.

---

> > > ### Author Response · Authors · 2025-11-27
> > >
> > > We thank the reviewer for the follow-up response. We are pleased that you acknowledge that most of your concerns have been addressed! We appreciate your position regarding the positive score.

---

### Official Review · Reviewer_EcpM · 2025-10-28

**Soundness:** 3
**Presentation:** 3
**Contribution:** 3
**Rating:** 4
**Confidence:** 5

**Summary:**

This paper proposes GOT-Edit, the first framework that integrates geometry-grounded reasoning into generic object tracking without requiring explicit 3D inputs. By introducing constraints during online model editing to preserve learned semantic consistency, GOT-Edit prevents model degradation while incorporating geometric cues that are often overlooked by conventional 2D trackers. Through online model editing with a null-space constraint, the method adaptively fuses geometric information while retaining semantic knowledge, thereby achieving greater robustness under occlusion, clutter, and visual ambiguity. The proposed approach has been evaluated on multiple state-of-the-art (SOT) benchmarks and demonstrates competitive performance.

**Strengths:**

1. The introduction of a geometry-aware correspondence learning mechanism is interesting and effective in visual tracking.
2. The proposed approach has been evaluated on multiple state-of-the-art (SOT) benchmarks and demonstrates competitive performance.

**Weaknesses:**

- Compared with the baseline, VGGT introduces additional computational overhead. It is recommended that the authors include a speed and FLOPs comparison to better illustrate efficiency.
- The VGGT component continuously updates learned knowledge during training. How does the method address potential error accumulation in this iterative learning process?
- While obtaining geometric features directly from 2D data without using 3D inputs can reduce data collection costs, it raises concerns about whether accurate geometric representations can be captured in this manner.
- The experiments are conducted using only 3 frames, and the observed performance gains might primarily come from the multi-template setting rather than the proposed algorithmic innovation. Moreover, since the model aims to learn geometric knowledge from multiple frames, there is a concern that using only three frames may be insufficient to capture rich and reliable geometric representations.

**Questions:**

See Weaknesses.

---

> ### Author Response · Authors · 2025-11-24
>
> We thank the reviewer for highlighting the geometry-aware correspondence learning mechanism as an interesting and effective component for visual tracking. We are also grateful for the recognition that our proposed approach achieves competitive performance on strong object tracking benchmarks, which confirms the viability of our method compared to state-of-the-art trackers. We address each of your concerns below and have revised the paper accordingly.
>
> ---
>
>
> > ### W1: Compared with the baseline, VGGT introduces additional computational overhead. It is recommended that the authors include a speed and FLOPs comparison to better illustrate efficiency. (1/2)
>
> We appreciate the reviewer's attention to the computational cost of our proposed method. We acknowledge that incorporating the Visual Geometry Grounded Transformer (VGGT) backbone introduces additional computational overhead compared to purely 2D trackers. This overhead, however, is the necessary cost for enabling geometry-aware generic object tracking without relying on external 3D sensors.
>
> As requested, we provide a detailed comparison of the speed (per-frame runtime) and model complexity (FLOPs) for the core components in Table A. The input sequence length is 3 for VGGT and 1 for DINO, consistent with the inference setting during tracking. Please note that FLOPs are device-agnostic and numerical precision-agnostic.
>
>
> **Table A. Runtime and FLOPs breakdown for VGGT, DINO, and the tracker component.**
>
> | Frame Resolution | Metric        | VGGT | DINO | Tracker Excluding VGGT & DINO |
> |------------------|--------------|------|------|-------------------------------|
> | 252 × 252        | Runtime (ms) | 65.6 | 8.7  | 9.8                           |
> |                  | FLOPs (G)    | 1000 | 105  | 32                            |
> | 378 × 378        | Runtime (ms) | 91.9 | 17.6 | 17.9                          |
> |                  | FLOPs (G)    | 2253 | 251  | 73                            |
>
>
> ---
>
> [Response continued in the next post]

---

> ### Author Response · Authors · 2025-11-24
>
> > ### W1: Compared with the baseline, VGGT introduces additional computational overhead. It is recommended that the authors include a speed and FLOPs comparison to better illustrate efficiency. (2/2)
>
> To expedite inference, we use StreamVGGT [1] instead of VGGT for geometric feature extraction (Table B). Consequently, geometry–semantic knitting for GOT-Edit achieves only 56.2 ms per incoming frame during tracking (on an NVIDIA GeForce RTX 4090 with PyTorch 2.0.0 and CUDA 11.7).
>
>
> **Table B. Efficiency–accuracy for VGGT and StreamVGGT.**
>
> | Tracker      | Geometry Method | Runtime | LaSOT | AVisT | NfS |
> |--------------|-----------------|-----|-------|-------|-----|
> | GOT-Edit-252 | VGGT            | 84.1  | 73.8  | 62.0  | 70.2 |
> | GOT-Edit-252 | StreamVGGT      | 56.2  | 73.4  | 61.9  | 69.8 |
>
> Here are the key observations from Table A and Table B:
>
> 1. As shown in Table A, the primary computational overhead is dominated by geometric (VGGT) feature extraction. Our core contribution, the online model editing module in *Tracker Excluding VGGT & DINO Backbone*, is efficient, with a runtime of only 9.8 ms at a 252 × 252 frame resolution.
>
> 2. As shown in Table B, we use mixed precision (BFloat16) for VGGT to accelerate inference (and training). Furthermore, experiments in Table B demonstrate that optimized geometric variants and selective feature application (we set the memory cache to 3 and apply geometric information every 3 frames in our StreamVGGT variant) can significantly increase the speed (e.g., runtime is reduced by approximately 40\% when StreamVGGT[1] replaces VGGT, while competitive accuracy is maintained.
>
> 3. The computational cost is the mandatory price for integrating geometric reasoning, and is justified by the superior accuracy (Table 4 of the paper) and robustness (Table 5 of the paper).
>
>
>
> We have added a detailed study on StreamVGGT in the section *Analysis of Alternate Geometry Backbone Choices* in the appendix (following the references section of the main paper) to reflect these results.
>
> **References**
>
> [1] D. Zhuo et al. Streaming 4D Visual Geometry Transformer. arXiv:2507.11539, 2025.

---

> ### Author Response · Authors · 2025-11-24
>
> > ### W2: The VGGT component continuously updates learned knowledge during training. How does the method address potential error accumulation in this iterative learning process?
>
> We thank the reviewer for this insightful comment. In GOT-Edit, the risk of error accumulation is mitigated by an online null-space constraint imposed on geometry-aware updates.
>
> As shown in Eq. 9–10 of the paper, the semantic branch predicts semantic weights $W_{\mathrm{sem}}$, while the geometry branch produces a perturbation $\Delta$. This perturbation is projected into the null space of the semantic features through $P_{\mathrm{null}}$, which ensures that geometry-aware updates do not overwrite semantic knowledge. The semantic features we use correspond to the learned knowledge used in AlphaEdit, which is regarded as knowledge to be preserved.
>
>
> Moreover, the entire procedure for predicting a geometry-aware and semantic-preserving tracking model is **online**. At each training iteration, the tracker predicts a new geometry-aware and semantic-preserving tracking model for the current frame. As noted in paragraph 3 of the *Implementation Details*, each iteration independently samples a subsequence that contains two reference frames and one current frame, without reuse of features from earlier iterations. The prediction of the tracking model is supervised by ground-truth labels through the loss in Eq. 12. Because both semantic knowledge and geometric updates are computed online for each subsequence and are constrained by the null-space projection, these design choices mitigate the risk of error accumulation across frames.

---

> ### Author Response · Authors · 2025-11-24
>
> > ### W3: While obtaining geometric features directly from 2D data without using 3D inputs can reduce data collection costs, it raises concerns about whether accurate geometric representations can be captured in this manner.
>
> We acknowledge that 3D geometric features derived solely from 2D inputs may be imperfect. However, our experimental results (e.g., row (1) versus row (3) in Table 5 of the paper) demonstrate that using the geometry extractor with only three input frames during tracking already yields substantial performance improvements on attributes such as occlusion and clutter, which can benefit from geometric awareness. This observation aligns with recent computer vision research, such as VGGT, which shows that a single view or a few views can yield sufficiently accurate geometric features.
>
> On the other hand, semantic features remain the dominant factor for object tracking. We use geometry as auxiliary information to complement missing geometric cues in tracking. Specifically, we learn a spatial binary mask to weight the geometric contribution (Eq. 5 of the paper) before predicting and projecting the geometry tracking model into the null space of the semantic features.
> This ensures that the resultant tracker becomes geometry-aware while rigorously preserving its semantic capability. Consequently, the tracker leverages the complementary strengths of both semantic and geometric information, thereby reducing the impact of imperfect geometric features.

---

> ### Author Response · Authors · 2025-11-24
>
> > ### W4: The experiments are conducted using only 3 frames, and the observed performance gains might primarily come from the multi-template setting rather than the proposed algorithmic innovation. Moreover, since the model aims to learn geometric knowledge from multiple frames, there is a concern that using only three frames may be insufficient to capture rich and reliable geometric representations.
>
> We clarify that the performance gains of GOT-Edit arise from the proposed online semantic–geometry editing mechanism, not from the multi-template setting. This is evidenced by the fact that **both the baseline tracker and GOT-Edit use the same number of input frames during tracking, namely two reference frames and the current frame, for fair comparison**. Therefore, the multi-template setting remains matched and cannot, by itself, explain the performance gap.
>
> Prior work on VGGT has already shown that even a single or a few RGB frames can provide sufficiently accurate geometric features, and our design follows this few-frame regime to exclude irrelevant information. We further ablate the number of reference frames in Table C. As shown, GOT Edit with two reference frames already surpasses the baseline by a clear margin, indicating that the main improvement comes from the proposed semantic–geometry editing mechanism rather than from increasing the number of templates. Adding more historical frames brings additional benefits to long-term datasets, such as LaSOT and AVisT, but shows similar performance on datasets containing fast-motion scenarios, such as NfS.
>
> **Table C. Effect of reference frame number on tracking performance.**
>
> | Tracker        | Ref Frames | LaSOT | AVisT | NfS  |
> |:-------------- |:----------:|:-----:|:-----:|:----:|
> | Baseline-252   | 2          | 70.7  | 59.2  | 68.5 |
> | GOT-Edit-252   | 2          | 73.3  | 62.0  | 70.2 |
> | GOT-Edit-252   | 3          | 73.8  | 63.0  | 70.4 |
> | GOT-Edit-252   | 4          | 74.0  | 63.3  | 69.9 |
>
> - - -
>
> We value the reviewer’s comments and the opportunity to clarify our methodology and results. Please let us know if additional explanation is needed.

---

> > ### Comment · Reviewer_EcpM · 2025-11-28
> >
> > I appreciate the authors' reply. My main concerns have been resolved, and I would like to update my rating to 6 accordingly.

---

> ### Author Response · Authors · 2025-11-28
> **Thank You and Follow-up on Final Score.**
>
> Dear reviewer,
>
> We are pleased to hear that we have addressed your concern and  and that you are going to upgrade the score to positive!
>
> **We would sincerely appreciate it if you could submit the final review scores for our paper, if they have not yet been submitted. Thank you for your insightful review.**
>
> Best,
> Authors

---

### Official Review · Reviewer_SbSc · 2025-10-30

**Soundness:** 4
**Presentation:** 4
**Contribution:** 4
**Rating:** 8
**Confidence:** 5

**Summary:**

This paper proposes GOT-Edit, a geometry-aware framework for online model editing in 2D object tracking. The key innovation lies in dynamically updating the target model during tracking, guided by multi-geometry information. The framework integrates a geometry-aware representation module with an online editing mechanism that refines the target model incrementally based on geometric consistency.

**Strengths:**

1. Introducing 3D geometric cues into online model editing for object tracking is both innovative and practical, providing a new perspective for improving robustness against shape and viewpoint variations.
2. The paper is well-written, logically structured, and easy to follow.
3. Demonstrates improvements on several benchmarks and provides qualitative examples showing better model adaptability during long-term tracking.

**Weaknesses:**

1. Since online model editing is computationally non-trivial, reporting runtime comparisons with baselines would help clarify practical deployment feasibility.

**Questions:**

Please see the weaknesses.

---

> ### Author Response · Authors · 2025-11-24
>
> We thank the reviewer for highlighting the three main strengths of our work: the novelty and practicality of introducing 3D geometric cues into online model editing, the clear presentation, and the consistent improvements across multiple benchmarks, accompanied by qualitative evidence of better long-term adaptability. This positive assessment supports our claim that semantics-preserving online model editing with geometric guidance is a useful and effective direction for advancing generic object tracking. We address each of your concerns below and have revised the paper accordingly.
>
> ---
>
>
> > ### W1. Since online model editing is computationally non-trivial, reporting runtime comparisons with baselines would help clarify practical deployment feasibility.
>
> We thank the reviewer for this suggestion. We provide runtime comparisons among individual components of GOT-Edit (Table A) and runtime comparisons with baselines (Table B).
>
> Table A details the component-wise runtime (in milliseconds, ms) costs and percentages for GOT-Edit, clarifying the source of overhead. The online model editing module (Align and Fuse and Model Predictors) accounts for 9.1 ms, only about 10.8% of the total runtime for GOT-Edit (84.1 ms), whereas the geometry backbone dominates the runtime overhead (78%). Runtime is measured per input–output pass for components.
>
> **Table A. Component-wise runtime (in milliseconds, ms) breakdown of the tracker on GOT-Edit.**
>
> | Frame Resolution | VGGT | DINO | Align & Fuse | Model Predictors | Reg/Cls Decoders | Total |
> |:----------------:|:----:|:----:|:-------------:|:-----------------:|:-----------------:|:-----:|
> | 252 × 252        | 65.6 | 8.7  | 2.3           | 6.8               | 0.7               | 84.1  |
> | 378 × 378        | 91.9 | 17.6 | 2.7           | 14.5              | 0.7               | 127.4 |

---

> ### Author Response · Authors · 2025-11-24
>
> As requested, we provide comparisons of runtime (in milliseconds, ms) and performance (AUC) with key baseline trackers:
>
> **Table B. Efficiency–accuracy for VGGT and StreamVGGT with varying cache and update frequency.**
>
> | Tracker                   | Frame Resolution | Geometry | Semantic | Tracker Excluding VGGT & DINO | AVisT | NfS  | LaSOT |
> |:--------------------------|:----------------:|:--------:|:--------:|:-------------------------------:|:-----:|:----:|:------:|
> | ToMP-L (DINO)             | 252 × 252        | –        | 8.7      | 6.8                             | 59.2  | 68.5 | 70.7   |
> | GOT-Edit (DINO+VGGT)      | 252 × 252        | 65.6     | 8.7      | 9.8                             | 62.0  | 70.2 | 73.8   |
> | GOT-Edit (DINO+StreamVGGT)| 252 × 252        | 37.7     | 8.7      | 9.8                             | 61.9  | 69.8 | 73.4   |
> | —                         | —                | —        | —        | —                               | —     | —    | —      |
> | ToMP-L (DINO)             | 378 × 378        | –        | 17.6     | 16.1                            | 62.0  | 69.0 | 72.7   |
> | PiVOT (DINO+CLIP)         | 378 × 378        | –        | 96.9     | 17.5                            | 62.2  | 68.2 | 73.4   |
> | GOT-Edit (DINO+VGGT)      | 378 × 378        | 91.9     | 17.6     | 17.9                            | 64.5  | 71.1 | 75.0   |
> | GOT-Edit (DINO+StreamVGGT)| 378 × 378        | 36.9     | 17.6     | 17.9                            | 64.3  | 70.7 | 74.8   |
>
> To enhance speed performance, we compare the effect of replacing VGGT with StreamVGGT [1] for geometric feature extraction in GOT-Edit.
> Introducing the geometry branch yields consistent AUC improvements over baseline trackers across all evaluated datasets. Moreover, replacing VGGT with StreamVGGT accelerates the geometry backbone by approximately 74% with only a moderate impact on performance, demonstrating a strong strategy for improving the practical deployment feasibility.
>
> We have updated the section *Computational Cost Analysis* in the appendix (following the references section of the main paper) to reflect these results. A more detailed analysis of different geometry backbone variants is provided in the section *More Experiments* in the appendix.
>
> **References**
>
> [1] D. Zhuo et al. Streaming 4D Visual Geometry Transformer. arXiv:2507.11539, 2025.
>
> - - -
>
> We thank the reviewer for both the positive feedback and the detailed suggestions that strengthened our work. If any issues remain unclear, we will be happy to address them.

---

### Official Review · Reviewer_z7fB · 2025-10-30

**Soundness:** 3
**Presentation:** 2
**Contribution:** 3
**Rating:** 6
**Confidence:** 3

**Summary:**

This paper proposes GOT-EDIT, an innovative approach that integrates 3D geometric features into general object tracking tasks through online model editing. It primarily leverages the VGGT module, trained on large-scale 3D datasets, to extract 3D geometric information from a set of 2D images. By employing null-space constraints, it seamlessly combines 3D geometric information with 2D semantic information, enabling the model to adaptively fuse 3D geometric information while preserving the semantic knowledge learned by the tracker. Extensive experiments demonstrate GOT-EDIT's robustness and accuracy.

**Strengths:**

The authors propose an innovative method for integrating 2D semantic information with 3D geometric information, and conducts experimental validation using datasets from diverse scenarios.

**Weaknesses:**

It is recommended to incorporate additional visualization results across diverse scenarios to substantiate the method's generality. While online model editing has achieved performance gains, it may introduce increased algorithmic complexity and computational overhead, potentially impacting real-time performance—particularly with high-resolution video inputs. Furthermore, the section detailing the online model editing methodology lacks detailed formulaic steps for operations based on AlphaEdit, which may cause confusion.

**Questions:**

1. One innovation of this paper is that seamlessly integrating 3D geometric information with 2D semantic information can enhance model performance. However, the mechanism by which 3D geometric information influences 2D semantic information remains undeclared. It is recommended to add relevant content.
2. Why was the VGGT module chosen for extracting 3D geometric features? It is recommend to compare it with other commonly used modules for 3D geometric feature extraction and detailing its specific advantages.
3. The paper employs null-space constraints but does not discuss whether these constraints may cause overfitting and performance degradation while preserving semantic integrity. The authors are advised to address this issue.
4. The fonts in Equations 6 and 7 do not match the subsequent text, potentially causing confusion. A thorough review and correction of relevant content is recommended.
5. Figure 1 lacks explanations for the annotations, including the red solid and dashed boxes as well as the green box.

---

> ### Author Response · Authors · 2025-11-24
>
> We thank the reviewer for acknowledging the innovation in integrating 2D semantic features with 3D geometric information and for noting the use of diverse datasets in the evaluation. This feedback supports our view that joint semantic–geometric reasoning is a promising direction for robust tracking across varied scenarios. We address each of your concerns below and have revised the paper accordingly.
>
> ---
>
>
> > ### W1: It is recommended to incorporate additional visualization results across diverse scenarios to substantiate the method's generality.
>
> Thank you for your suggestion. We have relocated the visualisation results from the appendix to the main paper (Figure 4), as the revision of the rebuttal allows additional space. To demonstrate the robustness of GOT-Edit, we have updated the video appendix with the file titled `GOT-Edit_Adverse_senario.mp4`. The video presents diverse scenarios that substantiate the method's generality. Please refer to the zip file in the Supplementary Material section of this review platform.
>
> > ### W2: While online model editing has achieved performance gains, it may introduce increased algorithmic complexity and computational overhead, potentially impacting real-time performance particularly with high-resolution video inputs. (1/2)
>
> We acknowledge that the inclusion of the geometric stream introduces algorithmic complexity and computational overhead, primarily from the feature extraction (VGGT accounts for $\approx 75\%$ of the total cost, see Table A). This overhead, however, is the necessary cost to enable geometry-aware object tracking without relying on external 3D sensors for superior tracking robustness (e.g., row (1) vs. row (6) in Table 4 of the paper).
>
> First, we present a more detailed comparison of per-frame runtime in milliseconds (ms) for the core components in Table A. Note that the input sequence length is 3 for VGGT, whereas it is 1 for DINO, consistent with the inference setting during tracking.
>
> **Table A. Runtime breakdown for VGGT, DINO, and the tracker component at different resolutions.**
>
> | Frame Resolution | VGGT | DINO | Align & Fuse | Model Predictors | Reg/Cls Decoders |
> |:----------------:|:----:|:----:|:-------------:|:-----------------:|:-----------------:|
> | 252 × 252        | 65.6 | 8.7  | 2.3           | 6.8               | 0.7               |
> | 378 × 378        | 91.9 | 17.6 | 2.7           | 14.5              | 0.7               |
>
> ---
>
> [Response continued in the next post]

---

> ### Author Response · Authors · 2025-11-24
>
> > ### W2: While online model editing has achieved performance gains, it may introduce increased algorithmic complexity and computational overhead, potentially impacting real-time performance particularly with high-resolution video inputs. (2/2)
>
>
> While we found that the proposed online editing module (Align and Fuse and Model Predictors) is efficient in Table A, we further enhanced the speed performance of the geometry backbone by using StreamVGGT [1] to replace VGGT for geometric feature extraction, and we report the results in Table B.
>
> In Table B, `GlobalAttn FineTune` refers to using DoRA [2] to fine-tune the linear layers of the global attention layer in the geometry model, which is the key mechanism for handling cross-frame information. `MemCache` refers to the number of historical K/V caches used for tracking. `Frequency` denotes the frequency for geometric feature extraction. All experiments were conducted using an NVIDIA GeForce RTX 4090 with PyTorch 2.0.0 and CUDA 11.7. The DoRA rank is set to 16, and only 2.4 M parameters are fine-tuned.
>
> **Table B.  Efficiency in runtime (ms per frame) and accuracy (%) for VGGT and StreamVGGT with varying cache and update frequency.**
>
> | Tracker       | Geometry Method | GlobalAttn FineTune | Mem Cache | Frequency      | Runtime | LaSOT | AVisT | NfS  |
> |:-------------:|:---------------:|:-------------------:|:---------:|:--------------:|:-------:|:-----:|:-----:|:----:|
> | GOT-Edit-252  | VGGT            | -                   | -         | Every Frame    | 84.1    | 73.8  | 62.0  | 70.2 |
> | GOT-Edit-252  | StreamVGGT      | -                   | 1         | Every Frame    | 72.5    | 72.8  | 61.4  | 69.7 |
> | GOT-Edit-252  | StreamVGGT      | $\checkmark$ | 1         | Every Frame    | 72.9    | 73.5  | 61.6  | 70.0 |
> | GOT-Edit-252  | StreamVGGT      | $\checkmark$ | 2         | Every 2 Frames | 59.4    | 72.3  | 61.8  | 69.5 |
> | GOT-Edit-252  | StreamVGGT      | $\checkmark$ | 2         | Every 3 Frames | 53.9    | 72.7  | 62.0  | 69.2 |
> | GOT-Edit-252  | StreamVGGT      | $\checkmark$ | 3         | Every 2 Frames | 67.8    | 73.1  | 62.7  | 70.0 |
> | GOT-Edit-252  | StreamVGGT      | $\checkmark$ | 3         | Every 3 Frames | 56.2    | 73.4  | 61.9  | 69.8 |
> | —             | —               | —                   | —         | —              | —       | —     | —     | —    |
> | GOT-Edit-378  | VGGT            | -                   | -         | Every Frame    | 127.4   | 75.0  | 64.5  | 71.1 |
> | GOT-Edit-378  | StreamVGGT      | -                   | 2         | Every 2 Frames | 84.6    | 74.3  | 63.2  | 69.5 |
> | GOT-Edit-378  | StreamVGGT      | $\checkmark$ | 2         | Every 2 Frames | 84.0    | 74.9  | 64.1  | 70.9 |
> | GOT-Edit-378  | StreamVGGT      | $\checkmark$ | 2         | Every 3 Frames | 72.4    | 74.8  | 64.3  | 70.7 |
> | GOT-Edit-378  | StreamVGGT      | $\checkmark$ | 3         | Every 2 Frames | 92.1    | 74.8  | 63.2  | 71.2 |
> | GOT-Edit-378  | StreamVGGT      | $\checkmark$ | 3         | Every 3 Frames | 78.4    | 75.2  | 63.3  | 71.4 |
>
>
>
>
> Here are the key observations from Table A and Table B:
>
> 1. The primary computational overhead is dominated by geometric (VGGT) feature extraction. Our core contribution, the online model editing module in `Tracker Excluding VGGT & DINO Backbone`, is efficient, with a runtime of only $9.8$ ms at $252 \times 252$  frame resolution or $17.9$ ms at $378 \times 378$ resolution.
>
> 2. The computational cost is the mandatory price for integrating geometric reasoning, and is justified by the superior accuracy (Table 4 of the paper) and robustness (Table 5 of the paper).
>
> 3. The StreamVGGT variant configured with three cache memories (Mem Cache = 3) and a frequency of every three frames (e.g., resolution $252 \times 252$), or with two cache memories and a frequency of every two frames (e.g., resolution $378 \times 378$), performs comparably to the original VGGT variant while reducing runtime by approximately 40%.
>
> We have updated Section B, *Computational Cost Analysis*, and Section C, *Analysis of Alternate Geometry Backbone Choices*, in the appendix (following the references section of the main paper) to reflect these results.
>
>
> **References**
>
> [1] D. Zhuo et al. Streaming 4D Visual Geometry Transformer. arXiv:2507.11539, 2025.
>
> [2] S.-Y. Liu et al. DoRA: Weight-Decomposed Low-Rank Adaptation. In ICML, 2024.

---

> ### Author Response · Authors · 2025-11-24
>
> > ### W3: The section detailing the online model editing methodology lacks detailed formulaic steps for operations based on AlphaEdit, which may cause confusion.
>
>
> We thank the reviewer for this helpful comment.
> In the revised subsection *Online Model Editing* in the Method section,
> we provide more explicit formulaic steps for operations based on AlphaEdit.
> We clarify that feature standardization and regularization are crucial to obtain a stable online formulation and to mitigate rank deficiency of the feature correlation matrix,
> which frequently occurs during online object tracking and would otherwise prevent direct application of AlphaEdit.
> Specifically, we add Eqs.~11 and 12 and explicitly define the following three steps:
>
> - **Feature Standardization and Regularization:**
>   Construction of the regularized correlation matrix
>   $\mathbf{M} = \mathbf{Z}\mathbf{Z}^\top + \lambda \mathbf{I}$ to ensure stability.
>
> - **Null-Space Selection:**
>   Selection of eigenvectors $\mathit{U}_{\text{null}}$
>   corresponding to low-energy eigenvalues to isolate the non-informative subspace.
>
> - **Symmetrization:**
>   Explicit symmetrization step
>   $\mathit{P}_{\text{null}} \leftarrow \tfrac{1}{2}(\hat{\mathbf{P}} + \hat{\mathbf{P}}^\top)$
>   applied to mitigate numerical drift.

---

> ### Author Response · Authors · 2025-11-24
>
> > ### Q1. One innovation of this paper is that seamlessly integrating 3D geometric information with 2D semantic information can enhance model performance. However, the mechanism by which 3D geometric information influences 2D semantic information remains undeclared. It is recommended to add relevant content.
>
> The mechanism by which 3D geometric information influences 2D semantic information is first defined in Eq. 5 of the main paper for naive fusion without the null space constraint. After the fused features pass through the encoder in Eq. 6 and the decoder in Eq. 7, the localization head of the tracker is obtained via Eq. 8.
>
> The aforementioned fusion mechanism does not guarantee preservation of dominant semantic features, which are crucial for object tracking. We therefore apply the null space constraint for online tracking model editing. Specifically, the null space projector $P_{\text{null}}$ is applied to the encoded geometric weight $\Delta$ via Eq. 10. We project this geometric weight onto the null space of the semantic feature, resulting in $\Delta'$, which is integrated with the semantic weights $W_{\text{sem}}$ for prediction of the localization head of the tracker via Eq. 9.
> The projected integration explicitly preserves semantic knowledge during the fusion process and explains how 3D geometric information is knitted together with 2D semantic information in online tracking model editing.
>
> This question is related to Weakness 3 (W3), and we have revised the paper to further improve clarity as suggested.
>
>
> > ### Q2. Why was the VGGT module chosen for extracting 3D geometric features? It is recommended to compare it with other commonly used modules for 3D geometric feature extraction and detailing its specific advantages.
>
>
> VGGT is chosen because it jointly learns 3D quantities (camera pose, point maps, and depth) from a small number of RGB frames and exhibits strong generalization ability among existing geometry models, which aligns well with generic object tracking on 2D video streams without additional 3D sensors.
>
> To compare with other commonly used modules for 3D geometric feature extraction, we also evaluated StreamVGGT, an incremental variant of VGGT, as shown in Table B discussed earlier. The advantages of VGGT stem from its use of full cross-attention, which provides slightly stronger geometric performance. In contrast, StreamVGGT employs causal attention with a historical K/V cache, achieving a superior accuracy–cost tradeoff and higher processing speed.

---

> ### Author Response · Authors · 2025-11-24
>
> > ### Q3. The paper employs null-space constraints but does not discuss whether these constraints may cause overfitting and performance degradation while preserving semantic integrity. The authors are advised to address this issue.
>
> Null-space constraints in GOT-Edit help prevent overfitting and performance degradation (e.g., those shown in Table 5 of our paper) rather than cause them.
> When geometry weights are projected into the null space defined by semantic features, only the components of the geometric update that are orthogonal to the semantic span are retained.
> This operation preserves the dominant semantic features while limiting the degrees of freedom of the geometric update, which reduces the risk that noisy or imperfect geometric cues overfit and corrupt semantic knowledge.
>
>
> In addition, the semantic and geometric backbones, which provide generalizable representations, are frozen, and only lightweight model predictors and decoder heads (approximately 50M parameters) are trained, in contrast to trackers that fully train a backbone with hundreds of millions to billions of parameters.
>
>
> > ### Q4. The fonts in Equations 6 and 7 do not match those of the subsequent text, which may cause confusion. A thorough review and correction of relevant content is recommended.
>
> We thank the reviewer for bringing this issue to our attention. We have revised the notation and all related text to ensure consistency throughout the paper.
>
>
> > ### Q5. Figure 1 lacks explanations for the annotations, including the red solid and dashed boxes, as well as the green box.
>
> We thank the reviewer for bringing this to our attention. We have updated the caption of Figure 1 to explicitly define these visual elements.
>
> ---
>
> We hope that the above clarifications address your concerns. We thank the reviewer for the encouraging evaluations and insightful suggestions, which helped refine our research. We are happy to provide any further clarification if needed.

---

> > ### Comment · Reviewer_z7fB · 2025-11-26
> >
> > Thank you for the detailed clarifications. I acknowledge that I have read both the rebuttal and the reviews from other members of the Reviewers.
> >
> > While the paper demonstrates strong tracking performance, it essentially remains a combination of the VGGT [1] and the AlphaEdit [2]. This work addresses the engineering challenges of adapting both components for generic object tracking, **but it falls slightly short in terms of innovation.**
> >
> > In summary, I tend to maintain my initial socre.
> >
> > [1] VGGT: Visual Geometry Grounded Transformer, CVPR 2025.
> > [2] AlphaEdit: Null-Space Constrained Knowledge Editing for Language Models, ICLR 2025.

---

> > > ### Author Response · Authors · 2025-11-26
> > > **We appreciate.**
> > >
> > > Thank you for the follow-up and for maintaining a positive assessment of GOT-Edit! We appreciate your comments and emphasize that our method addresses key challenges in adapting VGGT and AlphaEdit for generic object tracking while achieving strong performance.
> > >
> > > Our tracker builds on recent progress in geometry models (VGGT) and model editing methods (AlphaEdit). The integrated geometry backbone, VGGT, is essential for geometry feature prediction, and AlphaEdit provides theoretical support for the editing constraint in our work.
> > > We extend offline model editing of AlphaEdit to an online setting, featuring cross-modality editing and perception for generic object tracking from 2D streaming video alone.
> > > Our online editing mitigates the rank deficiency of the feature correlation matrix during tracking. Under a null space constraint, we jointly predict semantic and geometric tracking models to obtain a geometry-aware and semantics-preserving tracker that updates incrementally, thereby alleviating the limitations of AlphaEdit regarding precomputed knowledge, single modality, and offline pipeline.
> > >
> > > Any further discussion or remaining concerns are very welcome.

---

### Author Response · Authors · 2025-12-02
**Rebuttal Summary for GOT-Edit (Submission 432)**

We sincerely thank all reviewers for their insightful and constructive comments, and we appreciate the committee's coordination throughout the review process.

During the rebuttal period, we provided thorough and detailed responses to all review comments, addressing the concerns of the reviewers and revising the paper accordingly.

During the discussion, reviewers noted that the key concerns had been addressed and conveyed a more favorable assessment:

Reviewer ukrA raised the rating from 2 to 6, and Reviewer EcpM raised the rating from 4 to 6, as reflected in the posted discussion.

The other three reviewers maintained their initial positive assessments, resulting in updated ratings:

**8 (SbSc), 6 (HoKn), 6 (z7fB), 6 (ukrA), and 6 (EcpM)**.

---

We summarize the primary strengths of our contribution, as acknowledged by the reviewers:

* **Pioneering Geometric Reasoning for Generic Object Tracking:** GOT-Edit is a pioneering tracker that integrates visual geometry knowledge into generic object tracking using only 2D streaming video, improving robustness to occlusion, clutter, and geometric variation while requiring no explicit 3D input or additional sensors during tracking. This setting aligns with human vision, in which observers infer 3D geometry from a 2D plane for visual tracking.

* **Online Editing Mechanism across Modalities:** We introduce an online model editing mechanism for adaptive integration of semantic and geometric cues. This is achieved by projecting geometric updates into the null space orthogonal to the semantic span, thereby constructing a geometry-aware, semantics-preserving tracking model for online adaptation, which enhances reliability in dynamic and challenging visual environments.

* **Extensive Evaluation and Validation across Benchmarks:** The proposed method conducts extensive experiments across several challenging benchmarks, demonstrating clear advantages under adverse visibility, partial occlusion, and background clutter, and achieving consistent improvements over strong 2D tracking baselines.

---

We summarize the main reviewer comments with our responses, as reviewers concluded that the concerns were addressed.

* **Computational Efficiency and Feasibility:**
    Reviewers z7fB, SbSc, HoKn, and EcpM questioned the computational overhead of the geometry backbone and online editing. We provided a runtime breakdown of the method's components, showing that the geometry backbone is the dominant cost, whereas the online editing module contributes a moderate share of the total computation. To address concerns about geometry feature extraction, we further proposed a solution that reduces latency by approximately 40% while maintaining similar accuracy.

* **Novelty, Distinction from Prior Work, and Robustness:**
    Reviewer ukrA requested a clearer distinction from AlphaEdit, RGB+X trackers, and ToMP. We expanded the discussion of these relationships and revised the manuscript to better highlight the unique contributions of the proposed online, cross-modality framework relative to these existing methods. We clarified that GOT-Edit is the first online, cross-modality model editing framework for generic object tracking that derives geometry directly from 2D streaming video and adaptively fuses it with semantics under an explicit preservation constraint. Beyond AlphaEdit, our method jointly predicts semantic and geometric tracking models in an online setting, mitigates rank deficiency in the feature-correlation matrix, and applies a stable null-space projection. This constrained online editing also helps mitigate potential error accumulation (EcpM).

- - -

With the main concerns addressed and a supportive consensus reached among reviewers, we hope the proposed paradigm represents meaningful progress for the field. We remain grateful for the reviewers' time and the committee's consideration of this submission.

Authors of Submission 432

---

### Meta-Review · Area_Chair_DZAv · 2026-01-06

**Summary:**

GOT-Edit proposes an online cross-modality model-editing tracker that injects geometry cues inferred from a pre-trained visual-geometry transformer while explicitly preserving semantic discrimination via null-space constrained updates, and the discussion converges on the paper’s clear novelty and consistent empirical gains under challenging tracking conditions

**Reviewer Concerns:**

The main concerns during discussion focused on (i) the computational overhead of the geometry backbone and (ii) whether iterative online updates could accumulate errors; the authors addressed (i) with a detailed runtime breakdown and an efficient StreamVGGT variant that cuts latency by 40% with comparable accuracy, and addressed (ii) by clarifying that geometry updates are projected into the null space of semantic features so they do not overwrite semantic knowledge.

**Reviewer Scores:**

The final sentiment is clearly on the accept side, with two initially skeptical reviewers explicitly increasing their ratings after rebuttal/discussion (2→6 and 4→6) and the overall updated score set being 8, 6, 6, 6, 6 (SbSc/HoKn/z7fB/ukrA/EcpM), indicating broad alignment that the key issues were resolved.

---

### Decision · Program_Chairs · 2026-01-26

Accept (Poster)